# Efficient Learning of Deep State Space Models via Importance Smoothing

**John-Joseph Brady** [1]  **Nikolas Nüsken** [2]  **Yunpeng Li** [1]

## Abstract

Latent state space systems are ubiquitous in statistical modelling, arising naturally when time series are observed through noisy measurements. However, training deep state space models (DSSMs) at scale remains difficult. Two largely distinct strategies have emerged for training DSSMs. The first, auto-encoding DSSMs, trains generative models by optimising a variational lower bound. The second backpropagates through the outputs of classical sequential Monte Carlo (SMC) algorithms. Such approaches can train DSSMs for both discriminative and generative tasks, but their inherently sequential forward passes scale poorly on modern hardware. We propose *parallel variational Monte Carlo* (PVMC), a new training method that bridges these paradigms and robustly trains DSSMs for both discriminative and generative tasks. Across a set of benchmark experiments, PVMC matches or exceeds state-of-the-art performance while training $10\times$ faster than the fastest competing SMC-based approach.

## 1. Introduction

State-space models (SSMs) provide a general framework for describing time-evolving systems. In an SSM, system dynamics are governed by an unobserved latent state evolving according to Markovian dynamics, while observations are generated as random variables conditioned on this state. Owing to their flexibility, SSMs have been widely applied across diverse domains including target tracking (Blom, 1985), option pricing (Heston, 1993), ecology (Patterson et al., 2017), meteorology (Clayton et al., 2013), and neuroscience (Panisky et al., 2013).

Parameter estimation in SSMs (Kantas et al., 2015) is a challenging problem. Classical approaches such as expectation–maximisation (Dempster et al., 1977) and Bayesian parameter estimation (Andrieu et al., 2010) are typically computationally tractable only for models with a restricted algebraic structure or low-dimensional parameter spaces. This limitation has motivated the development of gradient-based optimisation methods, particularly in settings where SSM components are parameterised by neural networks.

Deep state-space models (DSSMs), which parameterise transition and observation models with neural networks, are commonly trained using one of two paradigms. The first frames the SSM as a variational auto-encoder (VAE) (Kingma & Welling, 2014), using an encoder network that is independent of the latent dynamics (Krishnan et al., 2017; Rangapuram et al., 2018; Lin & Michailidis, 2024). While this approach enables scalable, fully parallel training, it is poorly suited to supervised or semi-supervised objectives and yields loose variational bounds.

The second paradigm is differentiable sequential Monte Carlo (DSMC) (Jonschkowski et al., 2018; Chen & Li, 2023), which approximates the latent posterior with an importance sample generated by particle filtering (Chopin & Papaspiliopoulos, 2020). DSMC naturally supports supervised losses (Jonschkowski et al., 2018) and can provide tighter variational bounds than VAE-based approaches (Maddison et al., 2017). However, its inherently sequential structure limits parallelism on modern hardware, substantially increasing training costs.

We propose *parallel variational Monte Carlo (PVMC)*, which bridges these two paradigms. Like VAE-DSSMs, PVMC avoids sequential proposal mechanisms of particle filtering, enabling efficient parallel execution. At the same time, like DSMC, it constructs an importance-weighted approximation to the marginal posterior over latent states, supporting supervised training and therefore allowing meaningful latent representations. Specifically, PVMC targets the marginal smoothing posterior defined as the marginal distribution of the latent state at each time-step conditional on the full observation sequence, rather than the online (filtering) posterior targeted by most DSMC methods. Furthermore, PVMC yields tighter variational bounds than standard VAE-based methods by accounting for every possible trajectory

[1]Centre for Oral, Clinical and Translational Sciences, King's College London, London, United Kingdom [2]Department of Mathematics, King's College London, London, United Kingdom. Correspondence to: John-Joseph Brady <John-Joseph.brady@kcl.ac.uk>.

*Proceedings of the 43$^{rd}$ International Conference on Machine Learning*, Seoul, South Korea. PMLR 306, 2026. Copyright 2026 by the author(s).

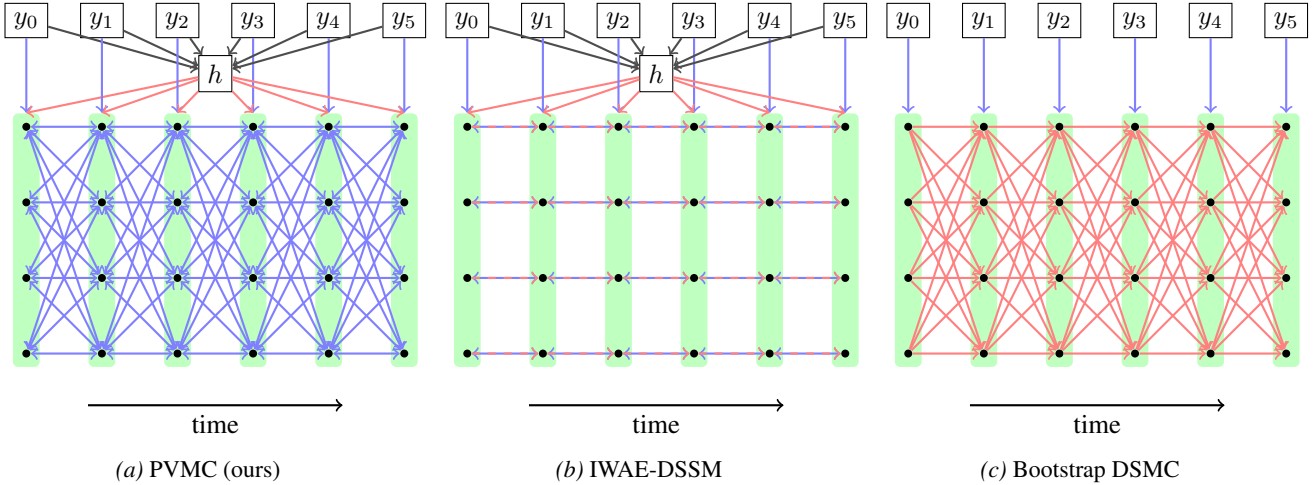

*(a)* PVMC (ours)       *(b)* IWAE-DSSM       *(c)* Bootstrap DSMC

*Figure 1.* Comparison of the sampling and weighting strategies. The black dots represent the sampled particles. Dependence via sampling and weighting are represented by red and blue arrows respectively. Red-blue dashed arrows show bi-directional dependence weighting and forward dependence via sampling. Grey arrows denote deterministic dependence. $h$ is a high-dimensional hidden variable that aggregates information over the complete trajectory of observations. DSMC is not amenable to parallelisation since, due to resampling, each particle's sampling distribution depends on all the particles at the previous time-step. IWAE-DSSMs are typically not used for state estimation tasks as they do not permit interaction between and therefore marginalisation over generated trajectories of particles.

through the set of proposed particles.

Our contributions are threefold:

- We introduce PVMC[*], the first end-to-end differentiable particle smoother with unbiased gradient updates and a statistically consistent posterior approximation.

- We derive a new evidence lower bound (ELBO) for training generative DSSMs that achieves a tighter variational bound than standard single-sample or importance-weighted objectives.

- We develop a hardware-aware recursion for the importance weights, achieving a $10\times$ speed-up over the fastest competing DSMC approach and $100\times$ over unbiased DSMC approaches in our experiments on an NVIDIA RTX 4090 GPU.

The paper is structured as follows. We review background on SSMs and parallel computation in Section 2. Section 3 presents PVMC and proves the correctness of the proposed recursions. Section 4 discusses related work. Experimental results are presented in Section 5. Section 6 concludes the paper.

## 2. Background

At a high level, PVMC will be constructed using importance-weighted approximations to the smoothing distribution un-

---

[*]Our implementation, including code to run the experiments, can be found at https://github.com/John-JoB/parallel-variational-sequential-monte-carlo

der a given SSM. The importance weights will be calculated via prefix/suffix scans, yielding an overall span complexity of $\mathcal{O}\left(\log N \times \log T\right)$. As a by-product of computing the importance weights, we will obtain an estimator of the log-likelihood with strong connections to the evidence lower bound of importance weighted auto-encoders (to be made specific in Section 3.2). In this section, we review relevant key concepts.

**State-space models:** We consider a time-discretised state-space model (SSM) with latent states $x_t \in \mathbb{R}^{d_x}$ and observations $y_t \in \mathbb{R}^{d_y}$:

$$x_0 \sim P\,, \tag{1}$$

$$x_{t>0} \sim M_t\left(\cdot \mid x_{t-1}\right)\,, \tag{2}$$

$$y_t \sim H_t\left(\cdot \mid x_t\right)\,. \tag{3}$$

Here, $P$ is the prior over the initial state, and $M_t$ and $H_t$ are the dynamic and observation kernels, respectively. All distributions considered may depend on the model parameters $\theta$, but we do not notate this explicitly for the purpose of readability. We assume that $P$, $M_t\left(\cdot \mid x_{t-1}\right)$, and $H_t\left(\cdot \mid x_t\right)$ admit densities for all values of $x_{0:T} \in \mathbb{R}^{(T+1)d_x}$. The joint smoothing density factorises as

$$Q_{0:T}\left(x_{0:T}\right) =$$
$$\frac{1}{L}P\left(x_0\right)\prod_{t=1}^{T} M_t\left(x_t \mid x_{t-1}\right)\prod_{t=0}^{T} H_t\left(y_t \mid x_t\right)\,, \tag{4}$$

where $L = p\left(y_{0:T}\right)$ is a normalisation constant.

In this paper we work with a parameterised family of SSMs and develop a class of novel algorithms for learning these parameters efficiently and effectively.

**Associative prefix sums:** It is well known that associative prefix sums (also known as scans) can be computed efficiently using parallel hardware (Blelloch, 1990). Given a sequence $a_{0:T}$, the prefix sum produces a new sequence $b_{0:T}$ where each element $b_t$ is an aggregation of the elements $a_{0:t}$. When the aggregation operator $\oplus$ is associative, $b_{0:T}$ admits a highly parallelised computation strategy.

Formally, let $(\mathbb{S}, \oplus)$ be a semigroup; with $a_{0:T} \in \mathbb{S}^{T+1}$. Then,
$$b_0 = a_0, \qquad b_{t>0} = b_{t-1} \oplus a_t. \qquad (5)$$

Define similarly the suffix sum which aggregates the elements $a_{0:T}$ in the backward direction:
$$\hat{b}_T = a_T, \qquad \hat{b}_{t<T} = a_t \oplus \hat{b}_{t+1}. \qquad (6)$$

We measure parallel efficiency of an algorithm via span complexity, i.e. the critical path length (Blumofe & Leiserson, 1999). Span complexity is the theoretical time complexity on hardware with an infinite number of processors. In Appendix D.2, we provide an efficient algorithm for computing the prefix and suffix sum in a single pass with span complexity $\mathcal{O}(\log T)$.

**Importance-weighted auto-encoders:** Variational auto-encoders (VAEs) (Kingma & Welling, 2014) are a paradigmatic approach to generative modelling. In general, VAEs model each datum $\tilde{y}$ as a possibly noisy observation of a random latent variable $\tilde{x}$:
$$\tilde{x} \sim \tilde{P}, \qquad \tilde{y} \sim \tilde{H}(\cdot \mid \tilde{x}). \qquad (7)$$

In the context of DSSMs, we can view an entire trajectory as one latent variable:
$$\tilde{x} = x_{0:T}, \quad \tilde{P}(\tilde{x}) = P(x_0) \prod_{t=1}^{T} M_t(x_t \mid x_{t-1}), \quad (8)$$

$$\tilde{y} = y_{0:T}, \qquad \tilde{H}(\tilde{y} \mid \tilde{x}) = \prod_{t=0}^{T} H_t(y_t \mid x_t). \qquad (9)$$

Given samples from a proposal distribution, $\tilde{X}^n \sim \tilde{V}(\cdot | \tilde{y})$, importance sampling yields an unbiased Monte Carlo estimator of the likelihood $p(\tilde{y})$:
$$\tilde{L}^N = \frac{1}{N} \sum_{n=1}^{N} w^n, \ w^n = \frac{\tilde{P}\left(\tilde{X}^n\right) \tilde{H}\left(\tilde{y} \mid \tilde{X}^n\right)}{\tilde{V}\left(\tilde{X}^n \mid \tilde{y}\right)}. \quad (10)$$

The optimal choice of $\tilde{V}(\cdot \mid \tilde{y})$, minimising the variance of $\tilde{L}^N$, is the posterior distribution of $\tilde{x} \mid \tilde{y}$ under the generative model (Equation (7)). We denote this posterior by $Q(\cdot \mid \tilde{y})$; typically, this distribution is intractable and one approximates it with a deep neural network.

Rather than maximising $\mathbb{E}\left[\tilde{L}^N\right]$, variational inference methods maximise the evidence lower bound (ELBO), $\mathbb{E}\left[\log \tilde{L}^N\right]$, which satisfies the standard IWAE monotonicity property (Burda et al., 2016),
$$\log p(\tilde{y}) \geq \mathbb{E}\left[\log \tilde{L}^{N+1}\right] \geq \mathbb{E}\left[\log \tilde{L}^N\right]. \qquad (11)$$

For standard VAEs, we have $N = 1$ and the ELBO decomposes as
$$\mathbb{E}\left[\log \tilde{L}^1\right] = \log p(\tilde{y}) - \mathcal{D}_{\text{KL}}\left(\tilde{V}(\cdot \mid \tilde{y}) \,\|\, Q(\cdot \mid \tilde{y})\right), \quad (12)$$

where $\mathcal{D}_{\text{KL}}$ denotes the Kullback-Leibler (KL) divergence. Maximising the ELBO in Equation (12) simultaneously optimises the likelihood and brings the proposal closer, in the KL-sense, to the posterior.

When $N > 1$, the objective corresponds to *importance weighted auto-encoding* (IWAE). Similarly to the standard case, $\mathbb{E}\left[\log \tilde{L}^N\right]$ is maximal only when $\tilde{V}(\cdot \mid \tilde{y}) \equiv Q(\cdot \mid \tilde{y})$ (Domke & Sheldon, 2018).

## 3. Parallel variational Monte Carlo

We focus on the time-marginal smoothing posteriors $Q_t(x_t \mid y_{0:T}) = p(x_t \mid y_{0:T})$ for a fixed time horizon $T$. For general nonlinear and non-Gaussian SSMs, these densities and their associated expectations are intractable, so we estimate them by Monte Carlo methods.

**Variational proposal:** We adopt a variational proposal over trajectories $x_{0:T} \sim V_{0:T}(\cdot \mid y_{0:T})$, suppressing (as usual) the neural network parameters in the notation. We assume that the proposal is fully factorisable across time:
$$V_{0:T}(x_{0:T} \mid y_{0:T}) = \prod_{t=0}^{T} V_t(x_t \mid y_{0:T}). \qquad (13)$$

At each time-step, we sample $N$ particles independently, $X_t^{1:N} \overset{\text{i.i.d.}}{\sim} V_t(\cdot \mid y_{0:T})$. We would like to emphasise that more general proposal distributions, such as structured inference models (Krishnan et al., 2017) as well as the proposal distributions associated to particle filters, can be used. However, deriving a well-defined importance sampler for these cases is more involved. We defer their discussion to Appendix E.2.

**Trajectory importance smoothing distribution:** For a given SSM, defined by Equations (1) to (3) and sequence of proposal distributions $V_t(\cdot \mid y_{0:T})$ as in Equation (13), the

importance weighted approximation Equation (10) of the joint smoothing distribution Equation (4) is given by

$$
Q_{0:T}^N \left( \cdot \mid y_{0:T} \right) = \frac{1}{\hat{L}^N N^{T+1}} \sum_{n_0, n_1, \ldots, n_T = 1}^N
$$

$$
\prod_{t=0}^T \left( K_t \left( X_t^{n_t}, X_{t-1}^{n_{t-1}} \right) \delta_{X_t^{n_t}} \right), \quad (14)
$$

with

$$
K_0 \left( X_0^{n_0}, \cdot \right) = P \left( X_0^{n_0} \right) \frac{H_0 \left( y_0 \mid X_0^{n_0} \right)}{V_0 \left( X_0^{n_0} \mid y_{0:T} \right)}, \quad (15)
$$

and

$$
K_{t>0} \left( X_t^{n_t}, X_{t-1}^{n_{t-1}} \right) =
$$

$$
M_t \left( X_t^{n_t} \mid X_{t-1}^{n_{t-1}} \right) \frac{H_t \left( y_t \mid X_t^{n_t} \right)}{V_t \left( X_t^{n_t} \mid y_{0:T} \right)}, \quad (16)
$$

where $\delta_{X_t^{n_t}}$ is the Dirac measure centred at $X_t^{n_t}$. Throughout, whenever $K_0$ appears in a product of the form $K_0 \left( X_0^{n_0}, X_{-1}^{n_{-1}} \right)$, we interpret it as $K_0 \left( X_0^{n_0}, \cdot \right)$. Thus, $X_{-1}$ and $n_{-1}$ are dummy quantities used only to keep the notation uniform. The smoothing distribution $Q_{0:T}^N$ in Equation (14) is an importance measure on every trajectory $x_{0:T}$ that can be formed by selecting one particle per time-step, yielding $N^{T+1}$ possible trajectories. Direct evaluation of the expectations with respect to $Q_{0:T}^N$ in Equation (14) is therefore computationally prohibitive.

### 3.1. The smoothing marginals

We now demonstrate how to represent the time-marginals of $Q_{0:T}^N \left( \cdot \mid y_{0:T} \right)$ as the result of an associative prefix sum so that they may be computed in $\mathcal{O} \left( \log N \times \log T \right)$ span complexity.

We time-marginalise $Q_{0:T}^N$ from Equation (14) to obtain

$$
Q_t^N \left( \cdot \mid y_{0:T} \right) = \frac{1}{\hat{L}^N N^{T+1}} \sum_{n_t = 1}^N w_t^{n_t} \delta_{X_t^{n_t}}, \quad (17)
$$

with weights

$$
w_t^{1:N} = \sum_{n_{-t} = 1}^N \prod_{u=0}^T K_u \left( X_u^{n_u}, X_{u-1}^{n_{u-1}} \right), \quad (18)
$$

and likelihood estimate

$$
\hat{L}^N = \sum_{n_0, n_1, \ldots, n_T = 1}^N \prod_{t=0}^T K_t \left( X_t^{n_t}, X_{t-1}^{n_{t-1}} \right), \quad (19)
$$

where $n_{-t}$ is the set of all particle indices except $n_t$. The supervised loss functions applied in differentiable particle

filtering, such as the mean square error (MSE) or the kernelised log likelihood (KNLL) (Jonschkowski et al., 2018), require the marginal smoothing distributions $Q_t^N$, or their moments, at every time-step. We compute $Q_t^N \left( \cdot \mid y_{0:T} \right)$ for each $t \in [0, \ldots, T]$ in parallel by combining the results from a prefix and a suffix sum.

The weights in Equation (18) are obtained by summing a product of local, time-adjacent factors over all particle indices except $n_t$. The standard sequential forward-backward recursion for chain-structured sums is sequential in $t$. The next theorem shows how to obtain all weights $w_t^i$ in $\mathcal{O} \left( \log N \times \log T \right)$ span by rewriting the required computation as associative prefix and suffix sums.

**Theorem 3.1.** *Assume, for demonstration, that $T$ is odd. Let $\mathbb{S} = \left\{ (C_1, C_2) \in \mathbb{R}^{N \times N} \times \mathbb{R}^{N \times N} \right\}$ be the set of tuples of $N \times N$ matrices, equipped with the associative operation*

$$
(C_1, C_2) \oplus (D_1, D_2) := (C_1, C_2 D_1 D_2). \quad (20)
$$

*Then the weights in Equation (18) can be computed as follows: Define the $\mathbb{S}$-valued sequence[†]*

$$
a_s = \left( \left\{ K_{2s} \left( X_{2s}^j, X_{2s-1}^i \right) \mid i, j \in \{1, \ldots, N\} \right\}, \right.
$$

$$
\left. \left\{ K_{2s+1} \left( X_{2s+1}^l, X_{2s}^k \right) \mid k, l \in \{1, \ldots, N\} \right\} \right), \quad (21)
$$

*for $s \in \left\{ 0, \ldots, \frac{T-1}{2} \right\}$. Let $b_s$ and $\hat{b}_s$ denote the prefix and suffix sum over $\{a_s\}$ under $\oplus$, as in Equations (5) and (6). Then, the weights $w_t^i$ can be extracted from $\{b_s, \hat{b}_s\}$ via:*

$$
w_t^i = \begin{cases}
\sum_{k,l,m}^N \omega_0^{1,i,k,l,m} & t = 0 \\
\sum_{k,l,m}^N \omega_{\frac{t-1}{2}}^{1,k,i,l,m} & t < T \text{ and odd,} \\
\sum_{k,l,m}^N \omega_{\frac{t}{2}-1}^{1,k,l,i,m} & t > 0 \text{ and even,} \\
\sum_{k,l,m}^N \omega_{\frac{T-3}{2}}^{1,k,l,m,i} & t = T,
\end{cases} \quad (22)
$$

*where*

$$
\omega_s^{i,j,k,l,m} = \left( (b_s)_1 \right)^{ij} \left( (b_s)_2 \right)^{jk} \left( \left( \hat{b}_{s+1} \right)_1 \right)^{kl} \left( \left( \hat{b}_{s+1} \right)_2 \right)^{lm}.
$$

*Proof.* See Appendix A. $\square$

It is straightforward to extend the algorithm illustrated in Theorem 3.1 to the situation where $T$ is even (see Appendix A). We provide detailed pseudocode for PVMC in Algorithms 1 and 2.

**Normalisation and consistency:** The normalisation constant $p \left( y_{0:T} \right) \approx \hat{L}^N$ can be estimated from the unnormalised importance weights via empirical normalisation:

$$
\hat{L}^N = \frac{1}{N^{T+1}} \sum_{n_t = 1}^N w_t^{n_t}, \quad (23)
$$

---

[†]Recall the notational convention introduced after Equation (16).

for any time-step $t$. The following results establish that the estimator Equation (17) converges at the standard Monte Carlo rate $\mathcal{O}_P\left(N^{-\frac{1}{2}}\right)$.

**Proposition 3.1.** *Assume that the importance ratios $K_0$ and $K_{t>0}$ introduced in Equations (15) and (16) are well posed; equivalently, the proposal densities are positive whenever the corresponding likelihood-weighted factors are nonzero. Then $\hat{L}^N$ is an unbiased estimator of the likelihood $p\left(y_{0:T}\right)$.*

*Proof.* See Appendix C.1. □

**Proposition 3.2.** *Assume the conditions of Proposition 3.1, and moreover that the importance weight of a single trajectory has finite variance: $Var\left[K_0\left(X_0,\cdot\right)\prod_{t=1}^{T}K_t\left(X_t,X_{t-1}\right)\right] < \infty$. Then, as $N \to \infty$, $\hat{L}^N$ converges in probability to $p\left(y_{0:T}\right)$ no slower than $N^{-\frac{1}{2}}$, that is,*

$$\hat{L}^N - p\left(y_{0:T}\right) = \mathcal{O}_P\left(N^{-\frac{1}{2}}\right). \tag{24}$$

*Proof.* See Appendix C.2. □

**Proposition 3.3.** *Under the conditions set out in Proposition 3.2, expectations with respect to the importance sampler in Equation (17) converge in probability to the expectation under the marginal smoothing posterior no slower than $N^{-\frac{1}{2}}$ as $N \to \infty$,*

$$\frac{1}{\hat{L}^N N^{T+1}}\sum_{n_t=1}^{N} w_t^{n_t} f\left(X_t^{n_t}\right) - \mathbb{E}_{x_t \sim SSM}\left[f(x_t) \mid y_{0:T}\right]$$
$$= \mathcal{O}_P\left(N^{-\frac{1}{2}}\right), \quad (25)$$

*for all bounded test functions $f$.*

*Proof.* See Appendix C.4. □

**Span complexity:** Multiplication of two $N \times N$ matrices is well known to have optimal span complexity $\mathcal{O}\left(\log N\right)$. The associative operator $\oplus$, defined by Equation (20), requires two such matrix multiplications. Therefore, the span complexity to compute the marginal importance weights Equation (18) from the kernels in Equations (15) and (16) is $\mathcal{O}\left(\log N \times \log T\right)$. In our experiments, the particles may be sampled and the kernels $K_t$ calculated in span complexity $\mathcal{O}\left(1\right)$, giving PVMC an overall forward span complexity of $\mathcal{O}\left(\log N \times \log T\right)$.

Calculating the gradients via back-propagation can be done by tracing the tree-like computation structure in reverse for the same $\mathcal{O}\left(\log T\right)$ depth. Each vector-Jacobian product in the tree recursion may be computed as a series of matrix multiplications so the overall span complexity of the backward pass is $\mathcal{O}\left(\log N \times \log T\right)$.

---

**Algorithm 1 PVMC**

1: **Input:** Time extent $T$; SSM $P$, $\{M_t\left(\cdot \mid x_{t-1}\right)\}_{t\in[1,\ldots,T]}, \{H_t\left(\cdot \mid x_t\right)\}_{t\in[0,\ldots,T]}$; proposal distributions $\{V_t\left(\cdot \mid y_{0:T}\right)\}_{t\in[0,\ldots,T]}$; observations $y_{0:T}$; number of particles $N$

2: **Output:** Particle locations $X_{0:T}^{1:N}$; weights $w_{0:T}^{1:N}$, likelihood estimate $\hat{L}^N$

3: **for** $n$ in $[1,\ldots,N]$ in Parallel **do**

4:  **for** $t$ in $[0,\ldots,T]$ in Parallel **do**

5:   Sample $X_t^n \sim V_t\left(\cdot \mid y_{0:T}\right)$

6:   $\left(W_V\right)_t^n \leftarrow V_t\left(X_t^n \mid y_{0:T}\right)$

7:   $\left(W_H\right)_t^n \leftarrow H_t\left(y_t \mid X_t^n\right)$

8:  **end for**

9:  $\left(W_P\right)^n = P\left(X_0^n\right)$

10: **end for**

11: **for** $n$ in $[1,\ldots,N]$ in Parallel **do**

12:  **for** $m$ in $[1,\ldots,N]$ in Parallel **do**

13:   **for** $t$ in $[1,\ldots,T]$ in Parallel **do**

14:    $\left(W_M\right)_t^{n,m} \leftarrow M_t\left(X_t^m \mid X_{t-1}^n\right)$

15:    $\left(W_K\right)_t^{n,m} \leftarrow \frac{\left(W_H\right)_t^n \left(W_M\right)_t^{n,m}}{\left(W_V\right)_t^n}$

16:   **end for**

17:   $\left(W_K\right)_0^{n,m} \leftarrow \frac{\left(W_H\right)_0^m \left(W_P\right)_0^m}{\left(W_V\right)_0^m}$

18:  **end for**

19: **end for**

20: **if** $T$ is even **then**

21:  (see Remark A.1 for details)

22:  $\left(\hat{W}_K\right)_0^{1:N,1:N} \leftarrow 1$

23:  $\left(\hat{W}_K\right)_{1:T+1}^{1:N,1:N} \leftarrow \left(W_K\right)_{0:T}^{1:N,1:N}$

24:  $S \leftarrow \frac{T}{2}$

25: **else**

26:  $\left(\hat{W}_K\right)_{0:T}^{1:N,1:N} \leftarrow \left(W_K\right)_{0:T}^{1:N,1:N}$

27:  $S \leftarrow \frac{T-1}{2}$

28: **end if**

29: **for** $s$ in $[0,\ldots,S]$ in Parallel **do**

30:  $a_s \leftarrow \left(\left(\hat{W}_K\right)_{2s}^{1:N,1:N}, \left(\hat{W}_K\right)_{2s+1}^{1:N,1:N}\right)$

31: **end for**

32: $W_{0:T}^{1:N} \leftarrow$ PVMC-Weights $\left(S, a_{0:S}, T, N\right)$ (Algorithm 2)

33: $\hat{L}^N = \frac{1}{N^{T+1}}\sum_{i=1}^{N} W_0^i$

34: **for** $n$ in $[1,\ldots,N]$ in Parallel **do**

35:  **for** $t$ in $[0,\ldots,T]$ in Parallel **do**

36:   $w_t^n \leftarrow \frac{1}{\hat{L}^N N^{T+1}} W_t^n$

37:  **end for**

38: **end for**

39: **Return:** $X_{0:T}^{1:N}$, $w_{0:T}^{1:N}$, $\hat{L}^N$

---

## 3.2. PVMC evidence lower bound

The interpretation of $\hat{L}^N$ as a likelihood estimator motivates a variational objective. We define the PVMC ELBO, $\mathcal{L}_{\text{PVMC}}^N = \mathbb{E}\left[\log \hat{L}^N\right]$. By Jensen's inequality we have $\mathcal{L}_{\text{PVMC}}^N \leq \log p\left(y_{0:T}\right)$. We may further define the following choices of ELBOs

$$\mathcal{L}_{\text{IWAE}}^N = \mathbb{E}\left[\log \frac{1}{N} \sum_{n=1}^{N} \prod_{t=0}^{T} K_u\left(X_t^n, X_{t-1}^n\right)\right], \quad (26)$$

$$\mathcal{L}_{\text{P-VAE}}^N = \mathbb{E}\left[\frac{1}{N^{T+1}} \sum_{n_0,\ldots,n_T=1}^{N} \log \prod_{t=0}^{T} K_t\left(X_t^{n_t}, X_{t-1}^{n_{t-1}}\right)\right], \quad (27)$$

$$\mathcal{L}_{\text{VAE}}^N = \mathbb{E}\left[\frac{1}{N} \sum_{n=1}^{N} \log \prod_{t=0}^{T} K_t\left(X_t^n, X_{t-1}^n\right)\right]. \quad (28)$$

**Theorem 3.2.** *The ELBOs defined in this paper obey the following hierarchies for $N \geq \tilde{N} \geq 1$:*

$$\log p\left(y_{0:T}\right) \geq \mathcal{L}_{PVMC}^N \geq \mathcal{L}_{IWAE}^N \geq \mathcal{L}_{IWAE}^{\tilde{N}} \geq \mathcal{L}_{P\text{-}VAE}^N = \mathcal{L}_{VAE}^N, \quad (29)$$

*and*

$$\mathcal{L}_{PVMC}^N \geq \mathcal{L}_{PVMC}^{\tilde{N}} \geq \mathcal{L}_{P\text{-}VAE}^N = \mathcal{L}_{VAE}^N. \quad (30)$$

*Proof.* See Appendix B. □

**Remark 3.1.** $\mathcal{L}_{IWAE}^N$ *and* $\mathcal{L}_{P\text{-}VAE}^N = \mathcal{L}_{VAE}^N$ *are the standard IWAE (Burda et al., 2016) and VAE (Kingma & Welling, 2014) ELBOs, respectively.*

## 4. Related work

As our method connects DSSMs and DSMC, it relates to both bodies of work. We review each in turn and highlight how PVMC inherits and differs from both paradigms.

Deep state space models (Lin & Michailidis, 2024) come from a rich and broad literature. DSSMs provide a natural probabilistic extension for recurrent neural network architectures (Chung et al., 2015). Recently, structured state space architectures such as S4 and Mamba (Gu et al., 2022; Gu & Dao, 2024) have garnered significant attention for their ability to capture long-term dependencies efficiently. However, these approaches forgo a probabilistic latent state, making them unsuitable for posterior approximation.

A common strategy for training DSSMs is to restrict part or all of the model to an algebraically tractable form, enabling partial analytical computation of the loss via Kalman filtering or smoothing (Krishnan et al., 2015; Johnson et al., 2016; Rangapuram et al., 2018). Calatrava et al. (2023) extend this approach to importance weighted objectives. For more general nonlinear and non-Gaussian DSSMs, the most common approach is to train on a variational objective equivalent to the standard VAE ELBO (Krishnan et al., 2017).

---

**Algorithm 2** PVMC-Weights

1: **Input:** Semigroup size $S$, semigroup elements $a_{0:S}$, time extent $T$, number of particles $N$
2: **Output:** Unnormalised particle weights $W_{0:T}^{1:N}$
3: $b_{0:S} \leftarrow$ Prefix-Sum $(a_{0:S}, \oplus)$;
  $\hat{b}_{0:S} \leftarrow$ Suffix-Sum $(a_{0:S}, \oplus)$; with $\oplus$ defined by Equation (20) (Algorithm 4 in Appendix D).
4: **for** $s$ in $[0\ldots, S-1]$ in Parallel **do**
5:   **for** $n$ in $[1,\ldots,N]$ in Parallel **do**
6:     **for** $m$ in $[1,\ldots,N]$ in Parallel **do**
7:       $l_s^n \leftarrow \sum_{i=1}^{N} \left((b_s)_2\right)^{i,n} \left((b_s)_1\right)^{1,i}$
8:       $r_s^m \leftarrow \sum_{i=1}^{N} \left(\left(\hat{b}_{s+1}\right)_2\right)^{m,i}$
9:       $c_s^{n,m} \leftarrow l_s^n r_s^m \left(\left(\hat{b}_{s+1}\right)_1\right)^{n,m}$
10:     **end for**
11:   **end for**
12:   **for** $n$ in $[1,\ldots,N]$ in Parallel **do**
13:     $\tilde{W}_{2s}^n = \sum_{i=1}^{N} c_s^{n,i}$
14:     $\tilde{W}_{2s+1}^n = \sum_{i=1}^{N} c_s^{i,n}$
15:   **end for**
16: **end for**
17: **if** $T$ is even **then**
18:   (see Remark A.1 for details)
19:   $W_{0:T-1}^{1:N} \leftarrow \tilde{W}_{0:T-1}^{1:N}$
20:   **for** $n$ in $[1,\ldots,N]$ in Parallel **do**
21:     $W_T^n \leftarrow \sum_{i=1}^{N} \left(\hat{b}_0\right)^{i,n}$
22:   **end for**
23: **else**
24:   $W_{1:T-1}^{1:N} \leftarrow \tilde{W}_{0:T-2}^{1:N}$
25:   **for** $n$ in $[1,\ldots,N]$ in Parallel **do**
26:     $W_T^n \leftarrow \sum_{i=1,j=1}^{N} \left((b_0)_1\right)^{i,j} \left(\left(\hat{b}_0\right)_2\right)^{j,n}$
27:     $W_0^n \leftarrow \sum_{i=1,j=1}^{N} \left((b_0)_1\right)^{i,n} \left(\left(\hat{b}_0\right)_2\right)^{n,j}$
28:   **end for**
29: **end if**
30: **Return:** $W_{0:T}^{1:N}$

---

Li et al. (2019) introduce a multiple sampling objective, however it remains equivalent in expectation to the single-sample VAE bound.

DSMC methods address some of these limitations by approximating latent posteriors using particle filters, naturally supporting supervised losses. However, the resampling step in classic particle filters involves sampling from a discrete distribution, and therefore breaks the reparameterisability of the algorithm. It has been shown experimentally that the direct application of reinforce to the resampling variables produces unstable gradients (Le et al., 2018), motivating a large body of work on effective estimators for the particle filtering gradients. Some approaches sacrifice unbiasedness

for lower variance (Karkus et al., 2018), while others introduce differentiable relaxations of resampling at the cost of substantial computational overhead (Corenflos et al., 2022; Li et al., 2024; Andersson & Zhao, 2025), even on highly parallel hardware.

Most DSMC methods target filtering distributions. To the best of our knowledge, the only closely related differentiable particle smoother is Mixture density particle smoother (MDPS) (Younis & Sudderth, 2024). Like the classical two-filter marginal smoother (Briers et al., 2010), it assimilates forward and backward particle filtering posteriors using a learned fusion network. While it can be effective in practice, the resulting marginals are not guaranteed to correspond to the smoothing distribution derived by the underlying DSSMs, and the method inherits biased gradient estimates from its constituent filters.

From a parallel computation standpoint, the primary bottleneck in SMC arises from resampling, which induces global dependencies between particles and requires thread synchronisation. Prior work has explored staggered resampling to alleviate this issue (Heine et al., 2014). Prefix summation has been leveraged to compute smoothing marginal for affine and Gaussian models Särkkä & García-Fernández (2021). This method was applied within the framework of Johnson et al. (2016) in Zhao & Linderman (2023) to learn Gaussian DSSMs. Trajectory-level smoothing for general SSMs is achieved in logarithmic span complexity by Corenflos et al. (2022). However, it requires repeated resampling and is therefore not suitable for end-to-end differentiation.

# 5. Experiments

We evaluate PVMC on (i) a linear-Gaussian system where exact smoothing is available; (ii) a nonlinear stochastic state estimation benchmark with supervised training and known latent states; and (iii) a real-world generative modelling task on financial time series. Full experimental details, architectures, and hyperparameters, are provided in Appendix G.

PVMC sits between differentiable SMC (DSMC) methods, and VAE-style DSSM training; no single family applies cleanly to all of our evaluation regimes, so we tailor baselines per experiment. Many classical particle smoothers are not end-to-end differentiable and are therefore unsuitable for training DSSMs; we use them only in the linear-Gaussian setting to verify that PVMC's forward pass behaves as a Bayesian smoother. For supervised non-linear state-estimation, we compare to DSMC baselines that support gradient-based training, namely the Soft (Karkus et al., 2018), Stop-gradient (Ścibior & Wood, 2021), and Diffusion (Andersson & Zhao, 2025) differentiable particle filters (DPFs) and the differentiable mixture density particle smoother (MDPS) (Younis & Sudderth, 2024). VAE-style

baselines are excluded since they do not provide a suitable mechanism for per-time-step marginal state estimation. For unsupervised generative modelling on SPX returns, we instead compare against widely used sequential generative baselines (DMM, TC-VAE) alongside Soft DPF, and omit methods that were unstable or exceeded our compute budget in this regime. Further baseline details are in Appendix G.2.

## 5.1. Linear Gaussian — state estimation

| Method | $e_x$ | Time (s) | KSD |
|---|---|---|---|
| RTS Smoother | 0.0 | 0.14 | — |
| Kalman Filter | 0.132 | 0.13 | — |
| TFS | 0.501 | 25.9 | 0.410 |
| d-SMC[‡] | 0.44 | 4.00 | 2.21 |
| PVMC (Kalman proposal) | 0.054 | 1.88 | 0.200 |
| PVMC (learned proposal) | 0.052 | 1.50 | 0.199 |

*Table 1.* Comparison of our model and a number of baseline approaches to the RTS smoother.

Firstly, we perform a comparison to the exact Rauch-Tung-Striebel (RTS) smoother (Rauch et al., 1965).

$$x_t = A x_{t-1} + q_t, \tag{31a}$$

$$y_t = H x_t + r_t, \tag{31b}$$

$$A \in \mathbb{R}^{d_x \times d_x}, \ A_{i,j} = 0.38^{|i-j|+1}, \tag{31c}$$

$$H \in \mathbb{R}^{d_y \times d_x}, \ H_{i,j} = \mathbb{1}\left(i = j\right), \tag{31d}$$

$$q_t \sim \mathcal{N}\left(0_{d_x}, I_{d_x \times d_x}\right), \tag{31e}$$

$$r_t \sim \mathcal{N}\left(0_{d_y}, I_{d_y \times d_y}\right), \tag{31f}$$

$$d_x = d_y = 5. \tag{31g}$$

This setting provides access to exact marginal posterior means via the RTS Smoother, allowing direct quantitative comparisons. We compare (i) the Kalman filter, (ii) the classical two-filter smoother (TFS) (Briers et al., 2010), (iii) the parallel-in-time smoother d-SMC (Corenflos et al., 2022), and (iv) PVMC in two variants: using the analytic Kalman filtering posterior as the proposal and using a learned neural proposal trained by maximising $\mathcal{L}_{\text{PVMC}}^N$. The latter evaluates the ability of PVMC to learn efficient proposal distributions, while all evaluated baseline approaches require the structural properties of the linear Gaussian model to derive an effective proposal distribution. Further experiment details and justification of our selection of baseline approaches can be found in Appendix G.3.

We report wall-clock time and the average L2 error between the estimated and exact RTS smoother posterior means. To demonstrate the methods' posterior fidelity we report a kernelised Stein discrepancy (KSD) (Liu et al., 2016) between

---

[‡]d-SMC targets the complete smoothing posterior, so a poor fit to the exact marginal posterior is expected.

| Method | MSE | Filtering MSE | 2-SWD | Time (m:s) | Failures |
|---|---|---|---|---|---|
| Stop-gradient | $0.83 \pm 0.50$ | $0.72 \pm 0.46$ | $14.8 \pm 9.4$ | $16{:}27 \pm 0{:}35$ | 2 |
| Soft | $0.62 \pm 0.42$ | $0.58 \pm 0.42$ | $6.70 \pm 4.30$ | $15{:}32 \pm 1{:}07$ | 7 |
| Diffusion | $0.52 \pm 0.22$ | $0.56 \pm 0.16$ | $10.2 \pm 4.28$ | $267{:}10 \pm 5{:}20$ | **0** |
| MDPS | $1.20 \pm 0.55$ | $1.32 \pm 0.64$ | $13.5 \pm 10.0$ | $26{:}23 \pm 0{:}14$ | 14 |
| P-VAE | $0.43 \pm 0.06$ | $1.21 \pm 0.11$ | $20.9 \pm 2.6$ | $\mathbf{1{:}49 \pm 0{:}01}$ | **0** |
| PVMC | $\mathbf{0.32 \pm 0.04}$ | $\mathbf{0.40 \pm 0.03}$ | $\mathbf{2.96 \pm 0.74}$ | $\mathbf{1{:}49 \pm 0{:}01}$ | **0** |

*Table 2.* The performance of our model and baseline approaches on the prey-predator state estimation task. Methods above the divider train via filtering, while methods below the divider train via smoothing. We report five metrics: the state estimation MSE of the learned algorithm; the state estimation MSE of a bootstrap particle filter (Gordon et al., 1993) using the learned SSM; the time to complete an entire training run; and the number of failed training runs out of the 20 repeats. All reported statistics are the sample means and standard deviations across the successful training runs.

the approximate posterior and the exact posterior, see Appendix G.3 for details. As shown in Table 1, PVMC achieves a significantly lower mean error than baselines compared with the RTS Smoother-generated exact marginal posterior means, while improving in runtime compared to TFS, the sequential particle-smoothing baseline. The learned proposal achieves the best result on all metrics evidencing that PVMC's ELBO gradients are an effective learning signal.

### 5.2. Prey-predator model — state estimation

We consider a stochastic variant of the Lotka-Volterra model (Wangersky, 1978) for the population density of a given species of prey and predator, denoted by $u$ and $v$, respectively, in the form used by (Andersson & Zhao, 2025):

$$\frac{du}{d\tau} = u\left(\tau\right)\left(\alpha - \gamma v\left(\tau\right) + \sigma dW_1\right), \qquad (32\text{a})$$

$$\frac{dv}{d\tau} = v\left(\tau\right)\left(\delta u\left(\tau\right) - \beta + \sigma dW_2\right), \qquad (32\text{b})$$

for $\alpha = \beta = 6$, $\gamma = 2$, $\delta = 4$, $\sigma = 0.15$ and $W_1, W_2$ are independent Brownian motions. The populations are observed at discrete intervals indexed by $t$ following a Poisson distribution that depends on the densities of the prey and predator species:

$$y_t \sim \text{Poisson}\left(\lambda\left(u_t, v_t\right)\right), \qquad (33\text{a})$$

$$\lambda\left(u_t, v_t\right) = \frac{5}{1 + \exp\left(\begin{bmatrix} -5u_t \\ -u_t v_t \end{bmatrix} + 4\right)}. \qquad (33\text{b})$$

We simulate data from this model in $\tau \in [0, 3]$ by Milstein's method and generate observations at $T = 256$ discretisation steps uniformly spaced in this range.

In this experiment we assume we know the characteristics of the observation method. However, the model dynamics are parameterised by a learned neural network. This experiment targets the supervised regime: we assume access to a set of independent reference trajectories endowed with ground-truth latent states. We train PVMC by minimising a linear

combination of the negative $\mathcal{L}_{\text{PVMC}}^N$ and the mean squared error between the estimated smoothing mean and the ground truth population densities.

We compare against three differentiable particle filter baselines, including the marginal variant of the stop-gradient filter (Stop-grad, Ścibior & Wood (2021)), the soft resampling filter (Soft, Karkus et al. (2018)), and the diffusion resampling filter (Diffusion, Andersson & Zhao (2025)), as well as the mixture density particle smoother (MDPS, Younis & Sudderth (2024)), the only other differentiable smoother available in the literature. We also include an ablative (P-VAE) that uses PVMC's importance sampler for state estimation but trains with $\mathcal{L}_{\text{P-VAE}}^N$, defined in Equation (27), instead of $\mathcal{L}_{\text{PVMC}}^N$. Full details and supplementary results can be found in Appendix G.4.

We evaluate each approach on its ability to learn to accurately predict the latent state. We also test the ability to train a DSSM that generalises to tasks not directly optimised for during training, namely the accuracy when used in classical particle filter and the discrepancy between the DSSM-implied filtering distribution and the true filtering distribution. Experimental results (Table 2 and Figure 6) show that PVMC achieves the best overall results on all tasks over 20 independent training runs, while exhibiting markedly improved training stability relative to DSMC baselines, which fail to converge reliably across seeds, matching the findings of Andersson & Zhao (2025).

### 5.3. Financial time series — generative modelling

We evaluate PVMC as a generative model on the Standard and Poor's 500 Index (SPX) daily close price. This setting reflects practical constraints where latent states are unobserved and datasets are limited or sensitive, motivating synthetic data generation to augment financial datasets for use in both optimising and back-testing business and investment strategies (Meldrum et al., 2025).

In this setting we do not assume the availability or existence

of a ground truth latent state. Instead, we interpret the latent state as a low-dimensional Markov variable summarising the state of the economy each day, mirroring the structure of classical financial models, e.g Hunt et al. (2000).

We assess whether generated trajectories reproduce key properties of the SPX returns: (i) volatility clustering via the autocorrelation of absolute and squared daily returns, (ii) distributional shape via skewness and kurtosis. We train on rolling windows of 120-day trajectory segments from the 10-year period between 2014-08-01 and 2024-08-01, by maximising $\mathcal{L}_{\text{PVMC}}^N$.

We compare against: time causal (TC) VAE (Acciaio et al., 2024); deep Markov model (DMM) (Krishnan et al., 2017), the soft resampling DPF (Karkus et al., 2018), along with the P-VAE ablation (which has the same architecture as PVMC but trained with the VAE ELBO objective as in DMM). Both the time causal and deep Markov model use the architectures proposed in the corresponding papers. TCVAE does not train a DSSM, so the PVMC architecture is not compatible with the TCVAE training methodology and vice-versa.

Figures 2 and 3 show that PVMC most consistently captures the short-term autocorrelations structure of SPX returns while also producing skewness and kurtosis distributions that are closer to those observed across real SPX slices. Neither DMM nor Soft-DPF were able to capture the correlation structure, suggesting they failed to learn a meaningful temporal dynamics. The empirical skewness and kurtosis are highly variable across different slices of the true SPX, making these distributional targets challenging to model. Nevertheless, PVMC provides the closest overall match, where TCVAE and P-VAE tend to under-approximate both the magnitude and spread of skewness and kurtosis. Soft-DPF and DMM again fail to capture these distributional properties. Further details and supplementary results are provided in Appendix G.5.

## 6. Conclusion

In this paper, we proposed a new training paradigm, named *parallel variational Monte Carlo* (PVMC), for learning deep state space models. PVMC combines parallel scalability with principled posterior inference. It avoids sequential re-sampling entirely by targeting the marginal smoothing posterior via importance weighting. This yields a statistically consistent smoother with unbiased gradient estimates, while enabling parallel-in-time computation through associative scan operations. We show that PVMC robustly trains deep state space models for both generative and discriminative tasks, achieving state-of-the-art accuracy in the simulated and real-world state-space modelling tasks and resulting in a $\sim 10\times$ speed-up compared to the fastest baseline approach for the evaluated supervised state estimation task.

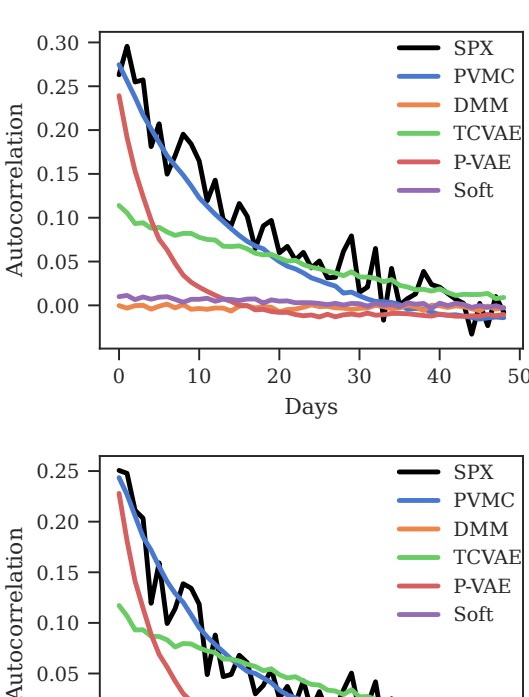

*Figure 2.* Comparison of the mean autocorrelation of absolute (above) and squared (below) daily returns over 1000 generated trajectories of 360 days to 6 non-overlapping real SPX trajectories of the same length.

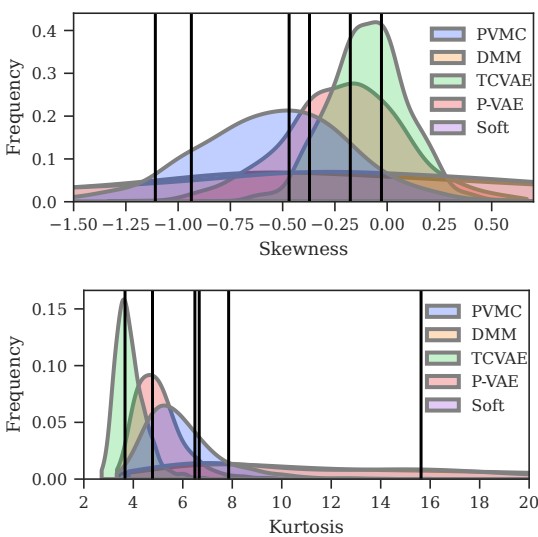

*Figure 3.* Comparison of the distribution of the skewness and kurtosis between the 1000 generated 360 day trajectories from each model. The black bars indicate the skewness and kurtosis for 6 non-overlapping 360-day trajectories from the real SPX.

## Acknowledgements

The authors thank the anonymous reviewers for their valuable input and feedback, which we have incorporated into this paper. JJ Brady acknowledges the support of an NPL/EPSRC DTP studentship.

## Impact statement

This paper presents work whose goal is to advance the field of machine learning. There are a wide variety of potential societal consequences of our work, none of which we feel must be specifically highlighted here.

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

# A. Proof of Theorem 3.1

**Lemma A.1.** $(\mathbb{S}, \oplus)$*, where $S$ is the set of tuples of $N \times N$ real matrices and $\oplus$ is defined in Equation* (20)*, is a semi-group.*

*Proof.* We first note that $c_1 \oplus c_2 \in \mathbb{S}$ for all $c_1, c_2 \in \mathbb{S}$, hence $(\mathbb{S}, \oplus)$ is closed. Furthermore, for $c_1, c_2, c_3 \in \mathbb{S}$, we have

$$(c_1 \oplus c_2) \oplus c_3 = ((c_1)_1, (c_1)_2(c_2)_1(c_2)_2) \oplus c_3 = ((c_1)_1, (c_1)_2(c_2)_1(c_2)_2(c_3)_1(c_3)_2),$$
$$c_1 \oplus (c_2 \oplus c_3) = c_1 \oplus ((c_2)_1, (c_2)_2(c_3)_1(c_3)_2) = ((c_1)_1, (c_1)_2(c_2)_1(c_2)_2(c_3)_1(c_3)_2),$$

by the associativity of matrix multiplication. Consequently, $\oplus$ is associative, and thus $(\mathbb{S}, \oplus)$ satisfies the semi-group axioms. $\qquad\square$

We note that each of the terms $K_t(\cdot, \cdot)$ is a function of two time-adjacent latent states. This independence structure permits the exchange of summation and multiplication over time-scales longer than 2.

**Lemma A.2.** *Let $A_i, \ldots, A_j \in \mathbb{R}^{N \times N}$ be square matrices. Then*

$$\left( \sum_{n_i, \ldots, n_{j-1}=1}^{N} \prod_{t=i}^{j} A_t^{n_t, n_{t-1}} \right)^{n_{i-1}, n_j} = \left( \bigotimes_{t=i}^{j} A_t \right)^{n_{i-1}, n_j}, \qquad i < j, \tag{34}$$

*where $\bigotimes$ is the order-preserving product of a sequence of matrices, and the upper indices refer to matrix entries.*

*Proof.* For convenience, we define $\bigotimes_{t=i}^{i} A_t \equiv A_i$. We proceed by induction: The base case is immediate, reducing to a single matrix product. Moreover,

$$\left( \sum_{n_i, \ldots, n_{j-1}=1}^{N} \prod_{t=i}^{j} A_t^{n_t, n_{t-1}} \right)^{n_{i-1}, n_j} = \sum_{n_{j-1}=1}^{N} \left[ \left( \sum_{n_i, \ldots, n_{j-2}=1}^{N} \prod_{t=i}^{j-1} A_t^{n_t, n_{t-1}} \right)^{n_{i-1}, n_{j-1}} A_j^{n_j, n_{j-1}} \right]$$
$$= \left( \left( \bigotimes_{t=i}^{j-1} A_t \right) A_j \right)^{n_{i-1}, n_j} = \left( \bigotimes_{t=i}^{j} A_t \right)^{n_{i-1}, n_j}, \tag{35}$$

where we have applied the inductive step in (35). Therefore, Equation (34) holds for all $j > i$. $\qquad\square$

We next apply Lemma A.2 to $a_{0:\frac{T-1}{2}}$, defined in Equation (21) for $0 < s < \frac{T-3}{2}$ and obtain

$$((b_s)_1)^{ij} = K_0 \left( X_0^j, \cdot \right), \tag{36}$$

$$((b_s)_2)^{jk} = \sum_{n_0, \ldots n_{2s}=1}^{N} \prod_{v=1}^{2s+1} K_v \left( X_v^{n_v}, X_{v-1}^{n_{v-1}} \right) \mathbb{1} \left( n_0 = j \right) \mathbb{1} \left( n_{2s+1} = k \right), \tag{37}$$

$$\left( \left( \hat{b}_{s+1} \right)_1 \right)^{kl} = K_{2s+2} \left( X_{2s+2}^l, X_{2s+1}^k \right), \tag{38}$$

$$\left( \left( \hat{b}_{s+1} \right)_2 \right)^{lm} = \sum_{n_{2s+2}, \ldots, n_T=1}^{N} \prod_{v=2s+3}^{T} K_v \left( X_v^{n_v}, X_{v-1}^{n_{v-1}} \right) \mathbb{1} \left( n_{2s+2} = l \right) \mathbb{1} \left( n_T = m \right), \tag{39}$$

where we recall the notational convention established after Equation (16).

Substituting Equations (36) to (39) into Equation (22) yields Equation (18). We demonstrate this explicitly case-by-case.

For $t = 0$, we have

$$\sum_{k,l,m=1}^{N} ((b_0)_1)^{1i} ((b_0)_2)^{ik} \left(\left(\hat{b}_1\right)_1\right)^{kl} \left(\left(\hat{b}_1\right)_2\right)^{lm} = \sum_{k,l,m=1}^{N} \left( K_0\left(X_0^i, \cdot\right) \times K_1\left(X_1^k, X_0^i\right) \times K_2\left(X_2^l, X_1^k\right) \right.$$

$$\times \sum_{n_3,\ldots,n_{T-1}=1}^{N} K_3\left(X_3^{n_3}, X_2^l\right) K_T\left(X_T^m, X_{t-1}^{n_{t-1}}\right) \prod_{v=3}^{T-1} K_v\left(X_v^{n_v}, X_{v-1}^{n_{v-1}}\right) \right) \quad (40)$$

$$= K_0\left(X_0^i, \cdot\right) \sum_{n_1,\ldots,n_T=1}^{N} \prod_{v=1}^{T} K_v\left(X_v^{n_v}, X_{v-1}^{n_{v-1}}\right) = w_0^i.$$

For $t < T$ and odd, we have

$$\sum_{k,l,m=1}^{N} \left(\left(b_{\frac{t-1}{2}}\right)\right)_1^{1i} \left(\left(b_{\frac{t-1}{2}}\right)_2\right)^{ik} \left(\left(\hat{b}_{\frac{t+1}{2}}\right)_1\right)^{kl} \left(\left(\hat{b}_{\frac{t+1}{2}}\right)_2\right)^{lm} = \sum_{k,l,m=1}^{N} \left( K_0\left(X_0^i, \cdot\right) \times K_{t+1}\left(X_{t+1}^l, X_t^i\right) \right.$$

$$\times \sum_{n_1,\ldots,n_t=1}^{N} K_1\left(X_1^{n_1}, X_0^k\right) \prod_{v=2}^{t} K_v\left(X_v^{n_v}, X_{v-1}^{n_{v-1}}\right) \mathbb{1}\left(n_t = i\right) \quad (41)$$

$$\times \sum_{n_{t+1},\ldots,n_T=1}^{N} \prod_{v=t+2}^{T} K_v\left(X_v^{n_v}, X_{v-1}^{n_{v-1}}\right) \mathbb{1}\left(n_{t+1} = l\right) \mathbb{1}\left(n_T = m\right) \right)$$

$$= \sum_{n_0,\ldots,n_T=1}^{N} \prod_{v=1}^{T} K_v\left(X_v^{n_v}, X_{v-1}^{n_{v-1}}\right) \mathbb{1}\left(n_t = i\right) = w_t^i.$$

For $t > 0$ and even, we have

$$\sum_{k,l,m=1}^{N} \left(\left(b_{\frac{t}{2}-1}\right)\right)_1^{1i} \left(\left(b_{\frac{t}{2}-1}\right)_2\right)^{ik} \left(\left(\hat{b}_{\frac{t}{2}}\right)_1\right)^{kl} \left(\left(\hat{b}_{\frac{t}{2}}\right)_2\right)^{lm} = \sum_{k,l,m=1}^{N} \left( K_0\left(X_0^k, \cdot\right) \times K_{t+1}\left(X_t^i, X_{t-1}^l\right) \right.$$

$$\times \sum_{n_1,\ldots,n_t=1}^{N} K_1\left(X_1^{n_1}, X_0^k\right) \prod_{v=2}^{t-1} K_v\left(X_v^{n_v}, X_{v-1}^{n_{v-1}}\right) \mathbb{1}\left(n_{t-1} = l\right) \quad (42)$$

$$\times \sum_{n_t,\ldots,n_T=1}^{N} \prod_{v=t+1}^{T} K_v\left(X_v^{n_v}, X_{v-1}^{n_{v-1}}\right) \mathbb{1}\left(n_t = i\right) \mathbb{1}\left(n_T = m\right) \right)$$

$$= \sum_{n_0,\ldots,n_T=1}^{N} \prod_{v=1}^{T} K_v\left(X_v^{n_v}, X_{v-1}^{n_{v-1}}\right) \mathbb{1}\left(n_t = i\right) = w_t^i.$$

For $T = t$, we have

$$\sum_{k,l,m=1}^{N} \left(\left(b_{\frac{T-3}{2}}\right)_1\right)^{1i} \left(\left(b_{\frac{T-3}{2}}\right)_2\right)^{ik} \left(\left(\hat{b}_{\frac{T-1}{2}}\right)_1\right)^{kl} \left(\left(\hat{b}_{\frac{T-1}{2}}\right)_2\right)^{lm} = \sum_{k,l,m=1}^{N} \left( K_0\left(X_0^k, \cdot\right) \times \right.$$

$$K_{T-1}\left(X_{T-1}^m, X_{T-2}^l\right) \times \sum_{n_1,\ldots,n_t=1}^{N} K_1\left(X_1^{n_1}, X_0^k\right) \prod_{v=2}^{T-2} K_v\left(X_v^{n_v}, X_{v-1}^{n_{v-1}}\right) \mathbb{1}\left(n_{T-2} = l\right) \times K_T\left(X_T^i, X_{T-1}^m\right) \right) \quad (43)$$

$$= \sum_{n_0,\ldots,n_{t-1}=1}^{N} K_T\left(X_T^i, X_{T-1}^{n_{T-1}}\right) \prod_{v=1}^{T} K_v\left(X_v^{n_v}, X_{v-1}^{n_{v-1}}\right) = w_T^i.$$

**Remark A.1.** *When $T$ is even, we can reduce to the odd case by prepending a dummy element, specifically the $N \times N$ all-ones matrix, to the sequence of kernels $K_{0:T}$, so that the resulting horizon is odd. Then the weights are obtained by running the same construction and then discarding the initial weights $w_0^{1:N}$, and finally re-indexing the remaining weights $w_{1:T+1}^{1:N}$ by shifting the time index down by one, i.e. $w_s^{1:N} \to w_{s-1}^{1:N}$.*

# B. Proof of Theorem 3.2

The proofs in this section are slightly modified variants of those given in Burda et al. (2016).

The first inequality

$$\log p\left(y_{0:T}\right) \geq \mathcal{L}_{\text{PVMC}}^N$$

is obtained from a direct application of Jensen's inequality.

The following lemma will prove useful:

**Lemma B.1.** *The total set of trajectories $\mathcal{T} = \{1, \ldots, N\}^{T+1}$ can be divided into $N^T$ disjoint subsets of $N$ elements each, where, within each subset, all trajectories are independently and identically distributed.*

*Proof.* We prove this lemma by constructing such a subset decomposition of trajectories. This can be conducted systematically as follows: take the set of trajectories where each particle has the same index, *i.e* $\{n_0, \ldots, n_T\} = n$. There are $N$ such trajectories. As no particles are repeated between any trajectory in this set, the trajectories are independent by the structure of our proposal distribution (Equation (13)). We designate this as the first of our $N^T$ subsets.

Now consider cyclic permutations of the indices at a given $t$ amongst the subset; there are $T + 1$ such positions to cycle. However, due to invariance under relabelling, cycling the first index leads to repeated trajectories. We therefore find $N^T$ subsets of $N$ independent terms. As we have considered permutations, it is guaranteed that no particle will be repeated within a subset. Consequently, the trajectories are independent. Furthermore, since our process guarantees that no trajectories are repeated and we have obtained the expected $N^{T+1}$ total, all distinct trajectories are accounted for. $\square$

**Remark B.1.** *We demonstrate the process from Lemma B.1 by writing down the $9$ subsets of $3$ trajectories obtained from the minimal example of $N = 3, T = 2$ below, where the rows of each subset represent the indices $(n_0, n_1, n_2)$ for each trajectory:*

$$\begin{bmatrix} 1,1,1 \\ 2,2,2 \\ 3,3,3 \end{bmatrix} \begin{bmatrix} 1,1,3 \\ 2,2,1 \\ 3,3,2 \end{bmatrix} \begin{bmatrix} 1,1,2 \\ 2,2,3 \\ 3,3,1 \end{bmatrix}$$
$$\begin{bmatrix} 1,3,1 \\ 2,1,2 \\ 3,2,3 \end{bmatrix} \begin{bmatrix} 1,3,3 \\ 2,1,1 \\ 3,2,2 \end{bmatrix} \begin{bmatrix} 1,3,2 \\ 2,1,3 \\ 3,2,1 \end{bmatrix} \quad (44)$$
$$\begin{bmatrix} 1,2,1 \\ 2,3,2 \\ 3,1,3 \end{bmatrix} \begin{bmatrix} 1,2,3 \\ 2,3,1 \\ 3,1,2 \end{bmatrix} \begin{bmatrix} 1,2,2 \\ 2,3,3 \\ 3,1,1 \end{bmatrix} .$$

By forming the subsets from Lemma B.1 systematically, we may assign an index $g$ to each of them, writing $\mathcal{T}_g \subset \mathcal{T}$. Then, let $r^g$ be the average over the unnormalised importance weights of all trajectories in subset $g$,

$$r^g := \frac{1}{N} \sum_{(n_0, n_1, \ldots, n_T) \in \mathcal{T}_g} \prod_{t=0}^{T} K_t\left(X_t^{n_t}, X_{t-1}^{n_{t-1}}\right). \quad (45)$$

Let $g'$ be a random index chosen uniformly from $\{1, \ldots, N^T\}$. Using the independence decomposition from Lemma B.1, we see that

$$\mathcal{L}_{\text{PVMC}}^N = \mathbb{E}\left[\log \frac{1}{N^T} \sum_{g=1}^{N^T} r^g\right] = \mathbb{E}\left[\log \mathbb{E}_{g'}\left[r^{g'}\right]\right] \quad (46)$$

$$\geq \mathbb{E}\left[\mathbb{E}_{g'}\left[\log r^{g'}\right]\right] = \mathbb{E}\left[\log r^1\right] \equiv \mathcal{L}_{\text{IWAE}}^N, \quad (47)$$

where we have applied Jensen's inequality and the fact that the marginal distribution of $r^g$ does not depend on $g$.

For $N \geq \widetilde{N} \geq 1$ we recall the following result from Burda et al. (2016):

$$\mathcal{L}_{\text{IWAE}}^N \geq \mathcal{L}_{\text{IWAE}}^{\widetilde{N}} \geq \mathcal{L}_{\text{IWAE}}^1. \quad (48)$$

We also have

$$\mathcal{L}_{\text{IWAE}}^1 = \mathbb{E}\left[\log\left[K_0\left(X_0^1,\cdot\right)\prod_{t=1}^{T}K_t\left(X_t^1,X_{t-1}^1\right)\right]\right] = \mathbb{E}\left[\log K_0\left(X_0^1,\cdot\right)\right] + \sum_{t=1}^{T}\mathbb{E}\left[\log K_t\left(X_t^1,X_{t-1}^1\right)\right], \tag{49}$$

$$\mathcal{L}_{\text{VAE}}^N = \frac{1}{N}\sum_{n=1}^{N}\mathbb{E}\left[\log\left[K_0\left(X_0^n,\cdot\right)\prod_{t=1}^{T}K_t\left(X_t^n,X_{t-1}^n\right)\right]\right] = \mathcal{L}_{\text{IWAE}}^1, \tag{50}$$

$$\mathcal{L}_{\text{P-VAE}}^N = \frac{1}{N^{T+1}}\sum_{n_0,\ldots,n_T=1}^{N}\mathbb{E}\left[\log\left[K_0\left(X_0^{n_0},\cdot\right)\prod_{t=1}^{T}K_t\left(X_t^{n_t},X_{t-1}^{n_{t-1}}\right)\right]\right] = \mathcal{L}_{\text{IWAE}}^1. \tag{51}$$

Therefore, we obtain the hierarchy:

$$\log p\left(y_{0:T}\right) \geq \mathcal{L}_{\text{PVMC}}^N \geq \mathcal{L}_{\text{IWAE}}^N \geq \mathcal{L}_{\text{IWAE}}^{\widetilde{N}} \geq \mathcal{L}_{\text{P-VAE}}^N = \mathcal{L}_{\text{VAE}}^N. \tag{52}$$

To prove the monotonicity of $\mathcal{L}_{\text{PVMC}}^N$ in $N$, the following two lemmas are helpful:

**Lemma B.2.** *For any indexed sequence of real numbers $a^{1:M} \in \mathbb{R}^M$, let $\mathcal{S}^{\widetilde{M}}$ be a collection of $\widetilde{M}$-element subsets of $\{1,\ldots,M\}$ such that*

$$\sum_{\mathcal{I}\in\mathcal{S}^{\widetilde{M}}}\mathbb{1}\left(m\in\mathcal{I}\right) = \bar{M} \quad \forall m \in \{1,\ldots,M\}.$$

*Furthermore, let $\mathcal{I}^{\widetilde{M}}$ be a uniformly sampled element of $\mathcal{S}^{\widetilde{M}}$. Then*

$$\frac{1}{\widetilde{M}}\mathbb{E}\left[\sum_{i\in\mathcal{I}^{\widetilde{M}}}a^i\right] = \frac{1}{M}\sum_{i=1}^{M}a^i. \tag{53}$$

*Proof.* By direct computation, we see that

$$\mathbb{E}\left[\sum_{i\in\mathcal{I}^{\widetilde{M}}}a^i\right] = \frac{1}{\left|\mathcal{S}^{\widetilde{M}}\right|}\sum_{\mathcal{I}^{\widetilde{M}}\in\mathcal{S}^{\widetilde{M}}}\sum_{i\in\mathcal{I}^{\widetilde{M}}}a^i = \frac{\bar{M}}{\left|\mathcal{S}^{\widetilde{M}}\right|}\sum_{i=1}^{M}a^i. \tag{54}$$

We also trivially have

$$\widetilde{M}\left|\mathcal{S}^{\widetilde{M}}\right| = M\bar{M},$$

thus completing the proof. $\qquad\square$

**Lemma B.3.** *Let $R$ be the direct product of $T+1$ copies of the symmetric group on $\{1,\ldots,N\}$. Thus, each element $\widehat{R} = (\widehat{R}_0,\ldots,\widehat{R}_T) \in R$ consists of one permutation of the particle indices at each time-step. Then, under the factorised proposal in Equation* (13)*, the joint law of the particles $X_0^1$ is invariant under the action of any $\widehat{R} \in R$.*

*Proof.* The joint density is given by

$$V\left(X_{0:T}^{1:N}|y_{0:T}\right) = \prod_{i=1}^{N}\prod_{t=0}^{T}V_t\left(X_t^i|y_{0:T}\right).$$

Let $\widehat{R}_t\left(i\right)$ be the new index of the particle $X_t^i$ after relabelling, and $\widehat{R}\left(X_{0:T}^{1:T}\right)$ be the array of particles after relabelling. We then have

$$V\left(\widehat{R}\left(X_{0:T}^{1:N}\right)|y_{0:T}\right) = \prod_{i=1}^{N}\prod_{t=0}^{T}V_t\left(X_t^{\widehat{R}_t(i)}|y_{0:T}\right) = \prod_{i=1}^{N}\prod_{t=0}^{T}V_t\left(X_t^i|y_{0:T}\right);$$

hence the joint distribution is invariant under relabelling of the particle indices. $\qquad\square$

With those lemmas in place, we can establish monotonicity as follows: let $\widehat{R} \in R$ be a uniformly selected element from the relabelling group $R$. Furthermore, let $\mathcal{T}^{\widetilde{N}}$ be the subset of all trajectories in $\mathcal{T}$ where every index is smaller than $\widetilde{N}$, that is

$$\mathcal{T}^{\widetilde{N}} = \left\{ \widehat{\mathcal{T}} \in \mathcal{T} \mid (\widehat{\mathcal{T}})_t \leq \widetilde{N} \ \forall t \in \{0, \ldots, T\} \right\}.$$

Since the trajectory labelling is invariant under the permutation $\widehat{R}$ followed by selection, all trajectories appear an equal number of times in the collection of selections formed by considering every possible relabelling. We then note that $\left| \mathcal{T}^{\widetilde{N}} \right| = \widetilde{N}^{T+1}$, so we may apply Lemma B.2 as follows:

$$\mathcal{L}_{\text{PVMC}}^N \equiv \mathbb{E}\left[ \log \frac{1}{N^{T+1}} \sum_{(n_0, \ldots, n_T) \in \mathcal{T}} \prod_{t=0}^{T} K_t\left( X_t^{n_t}, X_{t-1}^{n_{t-1}} \right) \right] \tag{55}$$

$$= \mathbb{E}\left[ \log \frac{1}{\widetilde{N}^{T+1}} \mathbb{E}_{\widehat{R}} \left[ \sum_{(n_0, \ldots, n_T) \in \mathcal{T}^{\widetilde{N}}} \prod_{t=0}^{T} K_t\left( X_t^{\widehat{R}_t(n_t)}, X_{t-1}^{\widehat{R}_{t-1}(n_{t-1})} \right) \right] \right] \tag{56}$$

$$\geq \mathbb{E}\left[ \mathbb{E}_{\widehat{R}} \left[ \log \frac{1}{\widetilde{N}^{T+1}} \sum_{(n_0, \ldots, n_T) \in \mathcal{T}^{\widetilde{N}}} \prod_{t=0}^{T} K_t\left( X_t^{\widehat{R}_t(n_t)}, X_{t-1}^{\widehat{R}_{t-1}(n_{t-1})} \right) \right] \right], \tag{57}$$

$$= \mathbb{E}\left[ \log \frac{1}{\widetilde{N}^{T+1}} \sum_{(n_0, \ldots, n_T) \in \mathcal{T}^{\widetilde{N}}} \prod_{t=0}^{T} K_t\left( X_t^{n_t}, X_{t-1}^{n_{t-1}} \right) \right] \equiv \mathcal{L}_{\text{PVMC}}^{\widetilde{N}}, \tag{58}$$

where we have applied Jensen's inequality to obtain Equation (57), and the final step follows because the system is invariant under $\widehat{R}$ by Lemma B.3. Therefore,

$$\mathcal{L}_{\text{PVMC}}^N \geq \mathcal{L}_{\text{PVMC}}^{\widetilde{N}}. \tag{59}$$

Taking $\widetilde{N} = 1$ in the same monotonicity result gives

$$\mathcal{L}_{\text{PVMC}}^{\widetilde{N}} \geq \mathcal{L}_{\text{PVMC}}^1. \tag{60}$$

For a single particle, the PVMC estimator reduces to the standard single-sample trajectory estimator, so that

$$\mathcal{L}_{\text{PVMC}}^1 = \mathcal{L}_{\text{IWAE}}^1. \tag{61}$$

Moreover, as shown in Equations (50) and (51),

$$\mathcal{L}_{\text{IWAE}}^1 = \mathcal{L}_{\text{P-VAE}}^N = \mathcal{L}_{\text{VAE}}^N. \tag{62}$$

Combining these inequalities yields the second hierarchy,

$$\mathcal{L}_{\text{PVMC}}^N \geq \mathcal{L}_{\text{PVMC}}^{\widetilde{N}} \geq \mathcal{L}_{\text{P-VAE}}^N = \mathcal{L}_{\text{VAE}}^N. \tag{63}$$

# C. Proofs of the results in Section 3

## C.1. Proof of Proposition 3.1

We compute

$$
\mathbb{E}\left[\prod_{t=0}^{T} K_t\left(X_t, X_{t-1}\right)\right] = \int \frac{P\left(dX_0\right)}{V_0\left(X_0 \mid y_{0:T}\right)} H_0\left(y_0 \mid X_0\right) \prod_{t=1}^{T} \frac{P_t\left(dX_t \mid X_{t-1}\right)}{V_t\left(X_t \mid y_{0:T}\right)} H_t\left(y_t \mid X_t\right) V_{0:T}\left(X_{0:T} \mid y_{0:T}\right) \tag{64}
$$

$$
= \int P\left(dX_0\right) H_0\left(y_0 \mid X_0\right) \prod_{t=1}^{T} P_t\left(dX_t \mid X_{t-1}\right) H_t\left(y_t \mid X_t\right) = p\left(y_{0:T}\right). \tag{65}
$$

Therefore, the expected value of the term contributed by any of the trajectories summed over in Equation (19) is equal to the likelihood. As $\mathcal{L}_{\mathrm{PVMC}}^{N}$ is estimated from an empirical mean of these terms, it follows that this the estimator is unbiased.

## C.2. Proof of Proposition 3.2

According to Proposition 3.1, we have that

$$
\mathbb{E}\left[\hat{L}^N - p\left(y_{0:T}\right)\right] = 0, . \tag{66}
$$

for all $N > 0$. To establish a rate of convergence in mean squared error (MSE), it thus remains to bound the variance.

We again perform the subset decomposition according to Lemma B.1. For each subset $\mathcal{T}_g$, define

$$
r^g := \frac{1}{N} \sum_{(n_0, n_1, \ldots, n_T) \in \mathcal{T}_g} \prod_{t=0}^{T} K_t\left(X_t^{n_t}, X_{t-1}^{n_{t-1}}\right). \tag{67}
$$

Each individual term has expectation $p(y_{0:T})$ and variance $\mathrm{Var}\left[\hat{L}^1\right]$. Since $r^g$ in an average of $N$ independent such terms, we obtain $\mathbb{E}[r^g] = p(y_{0:T})$ and $\mathrm{Var}\left[r^g\right] = \frac{1}{N}\mathrm{Var}\left[\hat{L}^1\right]$. Since furthermore

$$
\hat{L}^N = \frac{1}{G} \sum_{g=1}^{G} r^g, \tag{68}
$$

we see that

$$
\mathrm{Var}\left[\hat{L}^N\right] = \mathrm{Var}\left[\frac{1}{G} \sum_{g=1}^{G} r^g\right] \leq \frac{1}{N}\mathrm{Var}\left[\hat{L}^1\right]. \tag{69}
$$

From the unbiasedness of $\hat{L}^N$, it follows that

$$
\mathbb{E}\left[\left(\hat{L}^N - p\left(y_{0:T}\right)\right)^2\right] \leq \frac{1}{N}\mathrm{Var}\left[\hat{L}^1\right]. \tag{70}
$$

Convergence in MSE implies convergence in probability, so we may conclude

$$
\hat{L}^N - p\left(y_{0:t}\right) = \mathcal{O}_{\mathrm{P}}\left(N^{-\frac{1}{2}}\right). \tag{71}
$$

**Remark C.1.** *We have established that $\hat{L}^N$ obeys at worst the usual Monte Carlo $\sqrt{N}$ convergence, but this does not imply that $\hat{L}^N$ is asymptotically normally distributed,* i.e. *we have not established a central limit theorem.*

## C.3. Almost sure convergence of $\hat{L}^N$

**Proposition C.1.** *We stated Proposition 3.2 in terms of convergence of mean squared error, as it facilitated the derivation of a rate of convergence. However, under the same conditions, $\hat{L}^N$ obeys the stronger almost sure mode of convergence.*

$$
\hat{L}^N \xrightarrow{a.s.} p\left(y_{0:T}\right). \tag{72}
$$

*Proof.* The proof is similar to that of Proposition 3.2. But instead of using the rate of convergence of $r^g$, we note

$$r^g \xrightarrow{\text{a.s.}} p(y_{0:T}), \tag{73}$$

by the strong law of large numbers. A simple application of the Cesàro mean gives the required result. $\square$

### C.4. Proof of Proposition 3.3

The proof is similar to that of Proposition 3.2. The marginal smoother in Equation (17) is obtained by marginalising the complete trajectory smoother in Equation (14). Therefore, for any bounded test function $f$ of $x_t$, its expectation under the marginal smoother is equal to the expectation of the trajectory functional $x_{0:T} \mapsto f(x_t)$ under the complete trajectory smoother. Thus, it is sufficient to analyse the corresponding self-normalised importance sampling estimator on the complete trajectory space.

Let

$$\mu_t(f) := \mathbb{E}_{x_t \sim \text{SSM}}\left[f(x_t) \mid y_{0:T}\right]. \tag{74}$$

The marginal smoother in Equation (17) is a direct marginalisation of the complete trajectory smoother in Equation (14). For any joint distribution, the expectation of a bounded function with respect to a marginal is equal to its expectation under the joint. Thus, we can work directly with the joint importance-weighted distribution.

Define the unnormalised numerator

$$\hat{A}_t^N := \frac{1}{N^{T+1}} \sum_{n_0,\dots,n_T=1}^{N} f\left(X_t^{n_t}\right) \prod_{s=0}^{T} K_s\left(X_s^{n_s}, X_{s-1}^{n_{s-1}}\right). \tag{75}$$

Using the definition of the marginal weights in Equation (18), the estimator in Equation (17) can be written as

$$\frac{1}{\hat{L}^N N^{T+1}} \sum_{n_t=1}^{N} w_t^{n_t} f\left(X_t^{n_t}\right) = \frac{\hat{A}_t^N}{\hat{L}^N}. \tag{76}$$

By the same change-of-measure calculation used in the proof of Proposition 3.1, we have

$$\mathbb{E}\left[f(X_t) \prod_{s=0}^{T} K_s\left(X_s, X_{s-1}\right)\right] = \int p(dx_{0:T} \mid y_{0:T}) \, p(y_{0:T}) \, f(x_t) = p(y_{0:T}) \mu_t(f). \tag{77}$$

Therefore, by Equations (75) and (77),

$$\mathbb{E}\left[\hat{A}_t^N\right] = p(y_{0:T}) \mu_t(f). \tag{78}$$

Again we perform the decomposition in Lemma B.1 into $N^T$ subsets of independent trajectories. Recall the notation $\mathcal{T}_g$ for the set of trajectories in subset $g$, and define

$$a_t^g := \frac{1}{N} \sum_{(n_0,\dots,n_T)\in\mathcal{T}_g} f\left(X_t^{n_t}\right) \prod_{s=0}^{T} K_s\left(X_s^{n_s}, X_{s-1}^{n_{s-1}}\right). \tag{79}$$

By the construction in Lemma B.1, the trajectories within each subset are independent. Since $f$ is bounded and the single-trajectory importance weight has finite variance by the assumptions of Proposition 3.2, there exists a finite constant $\sigma_f^2$, independent of $N$ and $g$, such that

$$\text{Var}\left[a_t^g\right] = \frac{\sigma_f^2}{N}. \tag{80}$$

Moreover, writing $G = N^T$, Equations (75) and (79) imply

$$\hat{A}_t^N = \frac{1}{G} \sum_{g=1}^{G} a_t^g. \tag{81}$$

By the same argument as in the proof of Proposition 3.2, namely that the variance is non-increasing under averaging random variables with common finite variance, Equations (80) and (81) give

$$\text{Var}\left[\hat{A}_t^N\right] \leq \frac{\sigma_f^2}{N}. \tag{82}$$

Together, Equations (78) and (82) imply

$$\hat{A}_t^N - p\left(y_{0:T}\right)\mu_t(f) = \mathcal{O}_{\mathrm{P}}\left(N^{-\frac{1}{2}}\right). \tag{83}$$

Using the established convergence in probability of the likelihood estimate from Proposition 3.2,

$$\hat{L}^N - p\left(y_{0:T}\right) = \mathcal{O}_{\mathrm{P}}\left(N^{-\frac{1}{2}}\right). \tag{84}$$

Since $p\left(y_{0:T}\right) > 0$, $\hat{L}^N$ is bounded away from zero in probability. Therefore, by Equations (76), (83) and (84),

$$
\begin{aligned}
\frac{\hat{A}_t^N}{\hat{L}^N} - \mu_t(f) &= \frac{\hat{A}_t^N - \mu_t(f)\hat{L}^N}{\hat{L}^N} \\
&= \frac{\hat{A}_t^N - p\left(y_{0:T}\right)\mu_t(f) - \mu_t(f)\left(\hat{L}^N - p\left(y_{0:T}\right)\right)}{\hat{L}^N} \\
&= \mathcal{O}_{\mathrm{P}}\left(N^{-\frac{1}{2}}\right).
\end{aligned}
\tag{85}
$$

This proves the result.

# D. Parallel Scan Pseudocode

In this section, we adopt the short hand $a_{i:j} \leftarrow b_{i:j}$ to denote the setting of all elements in vector $a$ with indices between $i$ and $j$ in parallel. All summations of $M$ elements are implemented with a $\mathcal{O}\left(\log M\right)$ reduction, for example the one given in Algorithm 3.

## D.1. Parallel reductions

---

**Algorithm 3** Parallel Reduce

---

1: **Input:** Input tensor $a_{1:T}$, and arbitrary binary associative operator $\oplus$.
2: **Output:** Aggregation, $a_1 \oplus \cdots \oplus a_T$.
3: $A_{1:T} \leftarrow a_{1:T}$
4: $S \leftarrow T$
5: **for** $i$ in $[1, ..., \text{Ceil}\left(\log_2 T\right)]$ **do**
6:    $S \leftarrow \text{Length}\left(A\right)$
7:    $R \leftarrow \text{Ceil}\left(\frac{S}{2}\right)$
8:    **for** $j$ in $[1, \ldots, \text{Floor}\left(\frac{S}{2}\right)]$ in parallel **do**
9:       $B_j \leftarrow A_{2j-1} \oplus A_{2j}$
10:    **end for**
11:    **if** $S$ is odd **then**
12:       $B_R \leftarrow A_S$
13:    **end if**
14:    $S \leftarrow R$
15:    $A_{1:S} \leftarrow B_{1:S}$
16: **end for**
17: **Return:** $A_1$

---

## D.2. Parallel prefix and suffix sums

---

**Algorithm 4** Parallel prefix and suffix sums

---

1: **Input:** Input tensor $a_{1:T}$, and arbitrary binary associative operator $\oplus$ with identity element $I$.
2: **Output:** Prefix sums, $b_{1:T}$, Suffix sums, $\hat{b}_{1:T}$.
3: $A_{1:T}^1 \leftarrow a_{1:T}$
4: $R \leftarrow 1$
5: $S \leftarrow T$
6: **while** $S > 2$ **do**
7: $\quad Q \leftarrow \text{Ceil}\left(\frac{S+1}{2}\right)$
8: $\quad$ **for** $i$ in $[1, \ldots, R]$ in Parallel **do**
9: $\qquad$ **for** $j$ in $[1, \ldots, \text{Floor}\left(\frac{S}{2}\right)]$ in Parallel **do**
10: $\qquad\quad B_j^i \leftarrow A_{2j-1}^i \oplus A_{2j}^i$
11: $\qquad$ **end for**
12: $\qquad$ **if** $S$ is odd **then**
13: $\qquad\quad B_Q^i \leftarrow A_S^i$
14: $\qquad$ **else**
15: $\qquad\quad B_Q^i \leftarrow I$
16: $\qquad$ **end if**
17: $\qquad C_1^i \leftarrow A_1^i$
18: $\qquad$ **for** $k$ in $[1, \ldots, \text{Floor}\left(\frac{S-1}{2}\right)]$ in Parallel **do**
19: $\qquad\quad C_{k+1}^i \leftarrow A_{2k}^i \oplus A_{2k+1}^i$
20: $\qquad$ **end for**
21: $\qquad$ **if** $S$ is even **then**
22: $\qquad\quad C_Q^i \leftarrow A_S^i$
23: $\qquad$ **end if**
24: $\quad$ **end for**
25: $\quad D_{1:Q}^{1:R} \leftarrow C_{1:Q}^{1:R}$
26: $\quad D_{1:Q}^{R+1:2R} \leftarrow B_{1:Q}^{1:R}$
27: $\quad R \leftarrow 2R$
28: $\quad S \leftarrow Q$
29: $\quad A_{1:S}^{1:R} \leftarrow D_{1:Q}^{1:R}$
30: **end while**
31: **Return:** $A_1^{1:T-1}$, $A_2^{1:T-1}$

---

Algorithm 4 is an efficient algorithm for the computation of all prefix and suffix sums, barring the complete reduction, by reusing intermediates. This is a crucial memory optimisation over the naïve two scan approach during training where all intermediates need to be retained for the backwards pass.

# E. Extensions

### E.1. Reductions for estimators of multiplicative functions

A multiplicative functional is one which admits the factorisation

$$f_{0:T}(x_{0:T}) = f_0(x_0) \prod_{t=1}^{T} f_t(x_t, x_{t-1}) . \tag{86}$$

From Equation (14) it is clear that

$$\hat{L}^N Q_{0:T}^N(f_{0:T}) = \frac{1}{N^{T+1}} \sum_{n_0,\dots,n_T}^{N} K_0(X_0^{n_0}, \cdot) f_0(X_0^{n_0}) \prod_{t=1}^{T} K_t(X_t^{n_t}, X_{t-1}^{n_{t-1}}) f_t(X_t^{n_t}, X_{t-1}^{n_{t-1}}) . \tag{87}$$

Two important special cases are $f_{0:T} \equiv 1$, which yields the likelihood estimator $\hat{L}^N$, and $f_{0:T}(x_{0:T}) = f(x_t)$, which yields the unnormalised marginal expectation $\hat{L}^N Q_t^N(f)$. Thus, the same parallel reduction can be used either to estimate the likelihood or to estimate expectations under a single smoothing marginal.

The right-hand side of Equation (87) may be computed with a parallel reduction (Algorithm 3) of span complexity $\mathcal{O}(\log T)$, and, as above, $\hat{L}^N$ may be calculated by taking $f_{0:T}(\cdot) = 1$. This reduction has a substantially smaller work and memory cost than prefix/suffix sums, leading to a speed-up over calculating the full set of importance weights. We therefore use this method to calculate $\hat{L}^N$ in experiments where we do not require access to the individual particles or importance weights (such as in the financial time series setting from Section 5.3). If it is required to compute expectations of marginal functions at every time-step we maintain that Algorithm 1 should be the default; if $f(x_t)$ is vector-valued, then computing the direct reduction for every time-step will use more memory than calculating the importance weights by Algorithm 1.

### E.2. Non-factorisable proposal distributions

It is common for VAE-DSSMs to use proposal distributions where the particles at each time-step are non-independent, such as the structured proposals introduced in (Krishnan et al., 2017). Furthermore, the proposal structure of the classical particle filter has that each particle at time $t$ is dependent on the set of particles at time $t - 1$.

Consider importance sampling on the extended space $x_{0:T}^{1:N}$, and let

$$X_{0:T}^{\widehat{\mathcal{T}}} = \left\{ X_0^{\widehat{\mathcal{T}_0}}, X_1^{\widehat{\mathcal{T}_1}}, \dots, X_T^{\widehat{\mathcal{T}_T}} \right\} \tag{88}$$

denote a single trajectory indexed by $\widehat{\mathcal{T}} \in \mathcal{T} = (\widehat{\mathcal{T}_0}, \dots, \widehat{\mathcal{T}_T})$. The corresponding importance weight may be written as

$$w^{\widehat{\mathcal{T}}} \propto \frac{P^-\left(X_{0:T}^{-\widehat{\mathcal{T}}} \mid X_{0:T}^{\widehat{\mathcal{T}}}, y_{0:T}\right)}{V\left(X_{0:T}^{1:N} \mid y_{0:T}\right)} P\left(X_0^{\widehat{\mathcal{T}_0}}\right) H_0\left(y_0 \mid X_0^{\widehat{\mathcal{T}_0}}\right) \prod_{t=1}^{T} M_t\left(X_t^{\widehat{\mathcal{T}_t}} \mid X_{t-1}^{\widehat{\mathcal{T}_{t-1}}}\right) H_t\left(y_t \mid X_t^{\widehat{\mathcal{T}_t}}\right) , \tag{89}$$

where $X_{0:T}^{-\widehat{\mathcal{T}}}$ is the set of all particle locations not belonging to the target trajectory and $P^-\left(\cdot \mid X_{0:T}^{\widehat{\mathcal{T}}}, y_{0:T}\right)$ is the artificial target of all particles $X_{0:T}^{-\widehat{\mathcal{T}}}$.

For an importance sampler to be well defined the proposal distribution, $V$, must dominate the target; otherwise the Radon-Nikodym derivative in Equation (89) does not exist. When the proposal is factorisable into a product of marginals (Equation (13)) then the proposal is path-independent and this condition is trivially satisfied assuming that each time-marginal of the proposal dominates the corresponding marginal smoothing posterior. However, general proposals introduce path-dependence and it must be ensured that the target path has non-zero probability under the proposal. To illustrate this issue, consider a state-causal proposal with conditionally-factorised density

$$V\left(X_{0:T}^{1:N} \mid y_{0:T}\right) = \prod_{n=1}^{N} \left(V_0\left(x_0^n \mid y_{0:T}\right) \prod_{t=1}^{T} V_t\left(x_t^n \mid \{x_s^n\}_{s<t}, y_{0:T}\right)\right) . \tag{90}$$

Such a proposal only assigns positive probability to trajectories that follow its dependence chain. If we only consider target trajectories that follow this dependence chain, *i.e.* the trajectories $\widehat{\mathcal{T}_t} = n \forall t \in [0, \dots, T]$, such as in (Krishnan et al., 2017)

then no problem arises. However, we wish to include every possible trajectory through the set of sampled particles in our set of targets.

To restore domination whilst preserving a state-causal proposal we consider the following Markovian proposal distribution

$$V\left(X_{0:T}^{1:N} \mid y_{0:T}\right) = \left(\prod_{n_0=1}^{N} V_0\left(X_0^{n_0} \mid y_{0:T}\right)\right) \prod_{t=1}^{T} \prod_{n_t=1}^{N} \frac{1}{N} \sum_{i=1}^{N} V_t\left(X_t^{n_t} \mid X_{t-1}^i, y_{0:T}\right), \tag{91}$$

which considers that each particle is drawn from a uniformly weighted mixture over the set of particles at the previous time-step. Now, under the mild assumption that $H_t\left(y_t \mid x\right) M_t\left(\cdot \mid x\right) \ll V_t\left(\cdot \mid x'\right)$ for all possible $x, x'$, the support of $V$ includes every trajectory under consideration. We make the choice

$$P\left(X_{0:T}^{-\widehat{\mathcal{T}}} \mid X_{0:T}^{\widehat{\mathcal{T}}}, y_{0:T}\right) = \prod_{n_0 \neq \mathcal{T}_0} V_0\left(X_0^{n_0} \mid y_{0:T}\right) \prod_{t=1}^{T} \prod_{n_t \neq \mathcal{T}_t} \frac{1}{N} \sum_{i=1}^{N} V_t\left(X_t^{n_t} \mid X_{t-1}^i, y_{0:T}\right). \tag{92}$$

Substituting Equation (92) into Equation (89), $P\left(X_{0:T}^{-\widehat{\mathcal{T}}} \mid X_{0:T}^{\widehat{\mathcal{T}}}, y_{0:T}\right)$ cancels, and we obtain the simplified weights:

$$w^{\widehat{\mathcal{T}}} \propto \frac{P\left(X_0^{\widehat{\mathcal{T}_0}}\right)}{V_0\left(X_0^{\widehat{\mathcal{T}_0}} \mid y_{0:T}\right)} H_0\left(y_0 \mid X_0^{\widehat{\mathcal{T}_0}}\right) \prod_{t=1}^{T} \frac{M_t\left(X_t^{\widehat{\mathcal{T}_t}} \mid X_{t-1}^{\widehat{\mathcal{T}_{t-1}}}\right) H_t\left(y_t \mid X_t^{\widehat{\mathcal{T}_t}}\right)}{\frac{1}{N} \sum_{i=1}^{N} V_t\left(X_t^{\widehat{\mathcal{T}_t}} \mid X_{t-1}^i, y_{0:T}\right)}. \tag{93}$$

Equation (93) defines a valid importance sampler with a causally dependent proposal. When used inside PVMC $\hat{L}^N$ becomes an average of these importance weights across, in general, correlated samples; as each term is unbiased their average is unbiased. Similarly expectations of functions with respect to the smoothing marginals become averages of ratios of unbiased terms. However, we do not state or prove convergence results for dependent proposals.

**Remark E.1.** *The generalised proposal (Equation (91)) admits the same symmetry in the joint law of the particles under the relabelling group $R$ (Lemma B.3) as the factorised proposal (Equation (13)). Consequently, sampling a particle from a uniformly weighted mixture over ancestors is equivalent in distribution to deterministically selecting a path, up to relabelling. The mixture representation is required to establish dominance and in the weighting of the particles, but not for the implementation of the sampler. This view of importance sampling is connected to the ideas presented in (Elvira et al., 2019). This remark justifies the use of structured inference networks (Krishnan et al., 2017) to parameterise the proposal distribution.*

**Remark E.2.** *We can further extend Equation (93) to weighted mixtures where the weights of each mixture component, $w_{1:T}^{1:N}$, are positive almost surely.*

$$w^{\widehat{\mathcal{T}}} \propto \frac{P\left(X_0^{\widehat{\mathcal{T}_0}}\right)}{V_0\left(X_0^{\widehat{\mathcal{T}_0}} \mid y_{0:T}\right)} H_0\left(y_0 \mid X_0^{\widehat{\mathcal{T}_0}}\right) \prod_{t=1}^{T} \frac{M_t\left(X_t^{\widehat{\mathcal{T}_t}} \mid X_{t-1}^{\widehat{\mathcal{T}_{t-1}}}\right) H_t\left(y_t \mid X_t^{\widehat{\mathcal{T}_t}}\right)}{\sum_{i=1}^{N} \tilde{w}_t^i V_t\left(X_t^{\widehat{\mathcal{T}_t}} \mid X_{t-1}^i, y_{0:T}\right)}. \tag{94}$$

*In doing so, we permit an algorithm that samples from a particle filter and re-weights by PVMC. Note, however, that introducing mixture weights breaks index invariance. So, Remark E.1 does not apply in this context. Moreover, one would need to choose an appropriate gradient estimator or relaxation for the mixture component selection should $\tilde{w}_t$ depend on learnable parameters.*

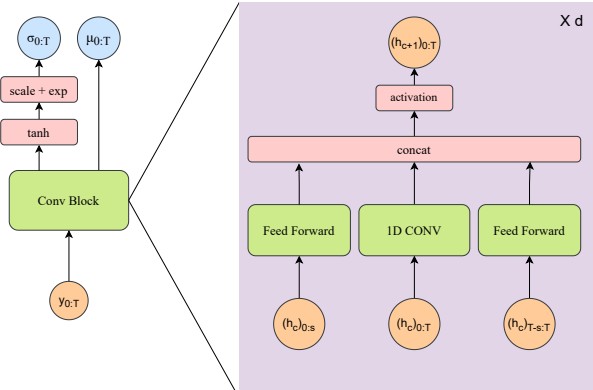

*Figure 4.* The proposal network we use in our experiments. The sampling distribution of the particles at time $t$ is a Gaussian with mean $\mu_t$ and diagonal covariance matrix $\text{diag}(\sigma_t \odot \sigma_t)$, where $\odot$ is the Hadamard (element-wise) product. The width of the convolutional kernel, the length of the input to the feed forward neural networks $s$, the depth $d$, the dimensionality of each hidden state $h_c$, and the choice of activation function are hyperparameters.

## F. Choice of proposal parameterisation

We do not aim to optimise the proposal network architecture in this paper, and therefore adopt simple, standard choices in our experiments. We nonetheless summarise common architecture choices that meet our assumptions.

In order to satisfy the condition of full factorisability (Equation (13)), it is required that the proposal distribution of each particle is conditionally independent from the realisation of every other particle. In practice, this can be implemented by using a sequence-to-sequence network to model the parameters of a reparameterisable distribution given the observations. Any sequence-to-sequence method such as a bidirectional LSTM (Hochreiter & Schmidhuber, 1997), transformer (Vaswani et al., 2017), or Mamba (Gu & Dao, 2024) is appropriate. In our experiments, we use a 1D convolution neural network that convolves over the time dimension, see Figure 4. For an odd convolution kernel width $\kappa$, the maximum temporal dependency is $d\frac{\kappa-1}{2}$, where $d$ is the number of stacked convolutional layers.

Markovian proposals, see Appendix E.2, also lead to unbiased estimation of the likelihood. However, they incur significantly higher work and memory cost as they require that the sampling density is calculated for each particle given every particle at the previous time-step.

# G. Experiment details

## G.1. General details

All experiments are performed with the same hardware and operating system:

- GPU: A single NVidia GeForce RTX 4090, 24GB of dedicated memory

- CPU: 24 core Intel Core i9-14900KF

- RAM: 64GB

- Operating System: Microsoft Windows 11 Enterprise 10.0.26100

All experiments are implemented in Python 3.12.11 and all tested approaches are implemented using PyTorch 2.8.0+cu129 (Paszke et al., 2019) as the numerical and auto-differentiation library. No implemented models make use of just in time (jit) compilation.

## G.2. Baselines

In total, we consider 1 ablation and 10 baseline approaches. Many baselines are not suitable for all the tasks that we wish to test PVMC on, so we vary the baselines between the experiments. We justify our choices individually for each experiment in Appendices G.3 to G.5

**Non-differentiable algorithms:**   These algorithms are not differentiable, so they cannot be used in the end-to-end training of DSSMs. In Section 5.1, we use them to verify that the forward pass of PVMC functions as a Bayesian smoother. In particular, we consider:

- Kalman filter (Kalman et al., 1960);

- Rauch-Tung-Striebel (RTS) Smoother (Rauch et al., 1965);

- Two filter smoother (TFS) (Briers et al., 2010);

- De-sequentialized Monte Carlo (d-SMC) (Corenflos et al., 2022).

Of these algorithms, the Kalman filter and the RTS smoother are exact algorithms and restricted to linear and Gaussian SSMs; TFS and d-SMC are particle methods that are suitable for use on a much more general class of non-linear SSMs. TFS is computationally sequential, whilst d-SMC, the Kalman filter, and the RTS smoother are implemented with span complexity $\mathcal{O}\left(\log T\right)$ recursions, for the latter two methods, due to our use of the algorithms presented in (Särkkä & García-Fernández, 2021). The RTS smoother and TFS calculate and approximate respectively the time-marginal smoothing distributions; d-SMC approximates the complete smoothing distribution and the Kalman filter calculates the filtering distribution.

**DSMC:**   DSMC algorithms can be applied to the same class of tasks as PVMC, including unsupervised training of generative models and the supervised training of generative and discriminative models. We consider:

- Soft differentiable particle filter (Soft DPF) (Karkus et al., 2018);

- Marginal stop-gradient differentiable particle filter (Stop-grad DPF) (Ścibior & Wood, 2021);

- Diffusion resampling differentiable particle filter (Diffusion DPF) (Andersson & Zhao, 2025);

- Mixture density particle smoother (MDPS) (Younis & Sudderth, 2024).

Of these algorithms, Soft DPF, Stop-grad DPF and Diffusion DPF approximate the filtering distribution on their forward pass; and MDPS approximates the marginal smoothing distribution. All three differentiable filters can be run with only a parameterisation of the SSM specified; the MDPS needs additional parameters to define a backwards-in-time proposal process and an assimilation rule to combine the forward and backward particles. The three filtering algorithms yield

statistically consistent estimators of expectations with respect to the filtering posterior under the SSM; however MDPS's assimilated weights are the output of a neural network, so we cannot give consistency guarantees on the output of its forward pass. Of the DSMC baselines considered, only Diffusion DPF yields unbiased estimates of either the weights or the particle locations with respect to its parameters.

**Variational Auto-Encoders:** The variational auto-encoders we consider do not provide mechanisms to marginalise over the latent states at each time-step and therefore target a distribution that grows in dimension exponentially with $T$. Therefore, these algorithms are not suitable for either state-estimation or supervised learning tasks. We consider:

- Deep Markov model (DMM) (Krishnan et al., 2017);

- Time-Causal variational auto-encoder (Acciaio et al., 2024).

For DMM, the generative component is a DSSM. For TCVAE, $p(x_{0:T})$ is parameterised by a normalising flow and the observations are the outputs of a sequence of deterministic functions $y_t = f_t(x_{0:t})$ that preserve causality. DMM targets a valid lower bound on the ELBO, equivalent in mean to $\mathcal{L}_{\text{VAE}}^N$, defined in Equation (28). For TCVAE, $p(y_t \mid X_{0:t})$ is zero almost everywhere. As a consequence, we cannot compute the weights by Equation (10); to rectify this, the conditional likelihood term is replaced by the L1 distance between $f_t(X_{0:t})$ and $y_t$.

### G.3. Linear Gaussian — state estimation

The aim of this experiment is to show that PVMC is effective at smoothing fully specified SSMs. Our choice to use a linear and Gaussian SSM means that we can compare directly to the exact marginal posterior. The true, data-generating SSM is the linear and Gaussian system used to benchmark DSMC algorithms in (Brady et al., 2025). For this experiment, we chose two non-differentiable smoothers to compare to, TFS and d-SMC. We exclude MDPS from this experiment as it is not capable of approximating the posterior of a specified SSM. TFS is a classic algorithm for approximating the marginal smoothing distribution at each time-step. d-SMC is, to the best of our knowledge, the only other parallel-in-time particle smoother. However, d-SMC targets the complete smoothing posterior, so we do not expect its output to be a good fit to the marginal posterior. As both of these particle smoothers use an analytically derived proposal distribution, we afford the same consideration to PVMC by proposing the particles from the exact filtering posteriors. To test the learning of an efficient proposal by optimising $\hat{L}^N$, and to more accurately mimic a realistic scenario where it is hard to derive an efficient proposal analytically, we also include a PVMC algorithm with a learned proposal. Finally we include the Kalman filter. Since we use the Kalman filter to propose particles for PVMC, comparison with the Kalman filter verifies that the PVMC weight recursion improves the estimate over taking uniformly weighted samples from the proposal.

For all particle methods, we propose 64 particles per time-step for 501 time-steps. We process in parallel 400 distinct testing trajectories in batches of 64, for all methods apart from PVMC, for which we take batches of 16. We adopt a different number of batches for PVMC because it is more compute intensive than other methods and incurs a significant slowdown when the GPU resources are stretched. The other methods run less simultaneous operations per trajectory than PVMC, so it is more efficient to process trajectories in a greater number of parallel batches. The total number of trajectories processed for each method remains constant. The mean square error is averaged over the 400 trajectories, each repeated 20 times. The reported elapsed times are the times in seconds to process the complete set of 400 trajectories, averaged over 20 repeats.

To assess the complete posterior fit we adopt a kernelised Stein discrepancy (KSD) between the particle approximation and the analytic posterior at the time-step $t = 249$ (Liu et al., 2016) with a radial basis function (Gaussian) kernel,

$$K(x, x') = e^{\frac{-\|x - x'\|_2^2}{2l^2}},$$

where the bandwidth $l$ is automatically selected. To select $l$ we simulate 400 independent observation trajectories $y_{0:T}$ and compute the density function $p(x_t \mid y_{0:T})$ for $t = 249$ by the RTS smoother. We then sample 64 particles from each posterior and calculate the squared median heuristic (Garreau et al., 2017) for each posterior. We take $l^2$ as the average of these squared median heuristics. We base the bandwidth $l$ on the target rather than the approximation, as is more usual, so that the bandwidth is constant across experiments. We choose to report the KSD in the main paper as it is a tractable and well-defined discrepancy between a weighted empirical distribution and a known Gaussian distribution. We supplement the main paper results with Table 3 which compares the 2-Wasserstein distance between the exact filtering posterior and the Gaussian obtained by moment matching to method's approximation. The 2-Wasserstein distance has an intuitive geometric

| Method | Kalman Filter | TFS | d-SMC | PVMC (Kalman proposal) | PVMC (Learned proposal) |
|---|---|---|---|---|---|
| 2 Wass. Dist. | 0.14 | 0.82 | 1.04 | 0.13 | 0.14 |

*Table 3.* Comparison of the approaches in their mean squared 2-Wasserstein distance from the approximate posterior to the exact posterior. For the Kalman filter the approximate posterior is the exact filtering posterior. For all other methods, a Gaussian is fitted to the weighted posterior by moment matching.

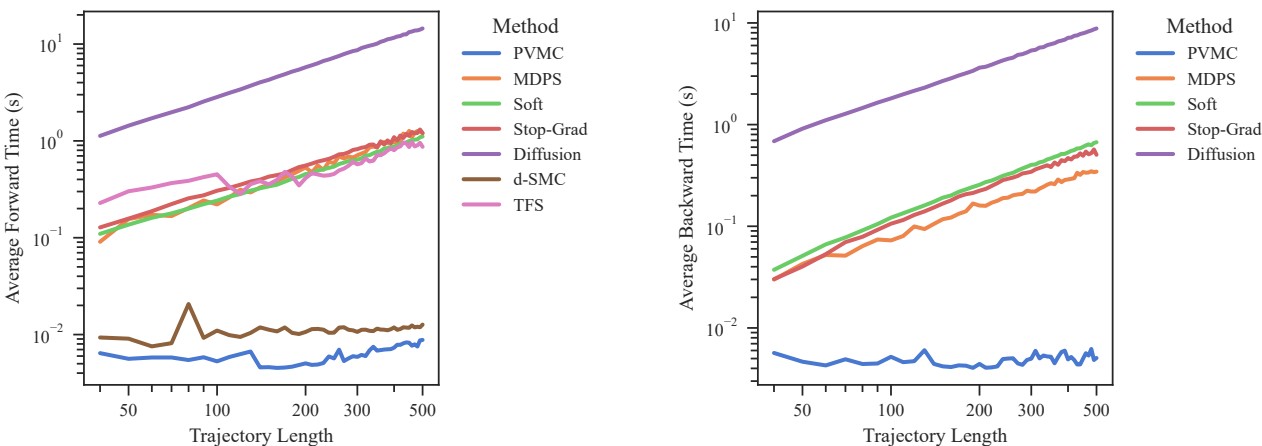

*Figure 5.* Comparison of runtimes of particle based methods for the Linear and Gaussian SSM specified in Equation (31).

interpretation. However, because we have employed moment matching it can only capture information about the first two moments of the approximation and does not capture any higher-order non-Gaussianity.

To parameterise the proposal distribution for the PVMC algorithm with a learned proposal, we use the proposal network described by Figure 4. We choose convolutional kernels of size 7, a hidden dimension of 16 channels per time-step, a selection length of $s = 9$ to input to the extremal feed-forward blocks, and stack $d = 4$ layers. Every layer uses a RELU activation, apart from the last one, which uses the identity function. The log standard deviation scaling factor is 1. The proposal has a total of $46,278$ parameters. We train our model with the ADAM optimiser with default parameters on the loss $\frac{-1}{T+1}\hat{L}^N$. During training we use a batch size of 32 and propose 32 particles per time-step for 501 time-steps. We train for 100 epochs on a total of 500 independent training trajectories. The final chosen model is the one which achieved the highest average $\hat{L}^N$ over a set of 100 validation trajectories.

d-SMC uses the Kalman filtering distributions to propose particles. The TFS proposes the backwards particles from the backwards Markov kernel implied by the forward SSM to ensure maximum cancellation of terms and therefore swift execution.

In addition to the experiment described above, with results in Table 1, we also measured the relationship between trajectory length and execution time for each of the particle methods. We separately report the forward pass and, for differentiable methods, backward pass runtimes. The PVMC and MDPS require additional parameters supplementary to the SSM – we train these parameters for 10 epochs to move away from any sparse or structured initialisations so that the execution times are representative of a realistic inference scenario. To facilitate fair comparison in this experiment, we propose particles for d-SMC from the same proposal as PVMC. Results are presented in Figure 5.

### G.4. Prey-predator mode — state estimation

The goal of this experiment is to demonstrate PVMC's ability to learn an SSM that accurately represents the distribution of the latent state by supervised training. The auto-encoding baselines DMM and TCVAE do not admit supervised losses so are excluded from this experiment, as are the non-differentiable methods. The data is simulated from a data-generating SSM given by Andersson & Zhao (2025). We choose this setting as, despite its simplicity, (Andersson & Zhao, 2025) found it challenging for DSMC algorithms with the highest number of non-diverging training runs from a grid search over hyper-parameters being 14 out of 20 and the best amongst their baselines being 9 out of 20.

Of interest is both that we can learn a smoother that performs well, but also that the learned SSM yields a meaningful representation of the latent state. The first objective is quantified by the mean squared error between the predicted latent state and the ground truth state. The second objective is assessed in two ways: firstly, we measure the mean squared error of a classical bootstrap particle filter with $N = 1,000$ particles using the SSM that we learned. Secondly, we calculate the squared empirical sliced 2-Wasserstein distance between importance weighted approximations of the posterior of the latent state at the last time-step under the learned SSM and under the true data-generating SSM. The squared sliced 2-Wasserstein distance is defined as the mean of the squared 2-Wasserstein distance of 512 random 1D projections of the particles.

Every method has the same SSM parameterisation. We follow Andersson & Zhao (2025) for the prior and observation model which have no learned parameters. Every trajectory is initialised with $x_0 = (2, 5)$. When sampling the prior we deterministically choose $x_0 = (2, 5)$, whilst when evaluating the prior's density we set the weights of every particle uniformly to $\frac{1}{N}$. The observation model is the true observation model Equation (33). Our dynamic model has the form

$$x_t = f_d\left(x_{t-1}\right) + x_{t-1} + \xi \odot e^{\tanh(f_\sigma(x_{t-1}))}, \qquad \xi \sim \mathcal{N}\left((0, 0), I_2\right), \tag{95}$$

where $\odot$ is the Hadamard (element-wise) product and $I_2$ is the $2 \times 2$ matrix identity, thus mimicking the form of an Euler–Maruyama integrator. The functions $f_d$ and $f_\sigma$ are fully-connected feed forward neural networks with 5 hidden layers of width 32. Every layer uses the swish activation function (Ramachandran et al., 2017) apart from the last one, which uses the identity function. The dynamic model has a total of $8,772$ trained parameters.

The PVMC and MDPS require additional parameterised components. To parameterise the proposal distribution for the PVMC algorithm, we use the proposal network described by Figure 4. We choose convolutional kernels of size 7 for all layers (apart from the last one, for which we use a kernel of size 5), a hidden dimension of 16 channels per time-step for the first hidden layer and 32 for all subsequent hidden layers, a selection length of $s = 9$ to input to the extremal feed-forward blocks, and stack $d = 5$ layers. In between each convolution layer we add a feed forward layer that acts across channels at single time-steps. Every layer uses a RELU activation (apart from the last one, which uses the identity function). The log standard deviation scaling factor is 1. The proposal has a total of $167,604$ parameters. The MDPS has a backwards SSM for which the prior is uniform over the range $[0, 5]$ for $(x_0)_1$ and $[0, 7]$ for $(x_0)_2$. The backwards observation model is the same as the forwards observation model. The backwards dynamical model uses a parameterisation of the same architecture as the forward model. The weight assimilation model follows the parameterisation described by Younis & Sudderth (2024) and has 513 total parameters.

We propose 32 particles per time-step for 257 time-steps and train in parallel batches of 16 trajectories. There are 100 trajectories in the training and testing datasets, and 50 in the validation dataset. We train each model for 100 epochs and return the model that achieved the lowest MSE on the validation dataset. We optimise $\left(\text{MSE} - \frac{1}{257}\text{ELBO}\right)$ using the Adam optimiser with default parameters.

For our ablation method, P-VAE, the MSE term in the loss is the PVMC MSE, however we use an ELBO equivalent in mean to $\mathcal{L}_{\text{VAE}}^N$. The low MSE achieved by the P-VAE, coupled with the poor Filtering MSE and 2-SWD, suggest that only the proposal component is being learned effectively for this model.

We visualise the results of our experiment in Figure 6. These plots demonstrate that whilst established DSMC methods such as Stop-Grad, Soft DPF and MDPS occasionally converge to a good representation, they suffer from much greater random seed dependence than PVMC.

### G.5. Financial time series — generative modelling

This final experiment is included to test the ability of PVMC to model complex sequential distributions. We chose to model the distribution of the returns of the SPX under the historical measure because it is a challenging problem in the following respects: data-availability is limited, we have only 2516 data-points all belonging to a single trajectory; and it is well established empirically that the distribution of asset returns is highly non-Gaussian (Cont, 2001). Furthermore, the distributions of the returns of financial assets are known to exhibit a number of stylised properties that are of interest to financial engineers which we can use to evaluate the quality of our simulated trajectories.

The considered baselines are the two auto-encoders, DMM and TCVAE, and Soft DPF. We exclude Diffusion DPF because it was too slow to run within our computation budget; Stop-grad DPF because it was unstable during training; and MDPS because it collapses to a simpler model that is very similar to the Stop-grad DPF when trained on unsupervised objectives. We also exclude all the non-differentiable baselines.

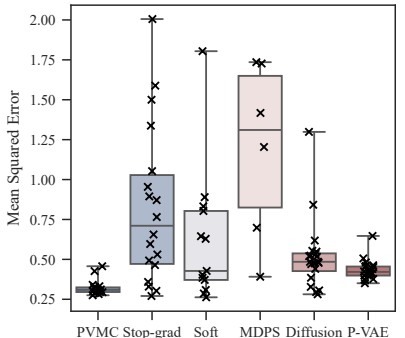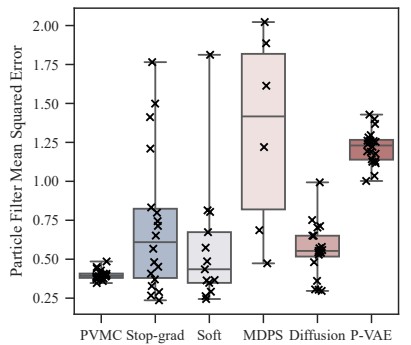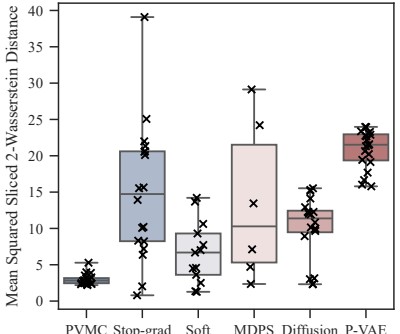

*Figure 6.* Box plots showing the spread of MSEs and squared sliced 2-Wasserstein distances achieved by each training approach for the different training runs for the prey-predator experiment. Failed runs are not plotted.

In this experiment we parameterise each considered model differently; for the DMM and TCVAE we adopt the proposal architectures of the papers that proposed them. For this reason we include an ablation, P-VAE, that trains the same architecture as PVMC, but using the DMM's variational target. We do not use the DMM proposal in a PVMC algorithm as it has a causal, non-factorisable proposal, see Appendix E.2.

We choose the dimension of the latent-state space to be 4. The prior model is a Student's t-distribution with 5 degrees of freedom, centred at zero and with a learnable scale parameter. The dynamic model is a conditional normalising flow with a Student's t-prior. We stack three conditional RealNVP (Dinh et al., 2017) layers, each with a hidden dimension of 32 (Chen & Li, 2024), with a total of $28,824$ learnable parameters. The observation model represents the distribution of log daily returns given the latent state as a mixture of Gaussians. All components of the mixture at all time-steps have the same learned variance, but the weights and locations of each mixture component are neural functions of the latent state. We also learn a constant temperature parameter that controls how uniform the weights tend to be. The total number of parameters for the observation model is $27,362$.

To parameterise the proposal distribution for the PVMC algorithm, we use the proposal network described by Figure 4. We choose convolutional kernels of size 7, a hidden dimension of 16 channels per time-step, a selection length of $s = 9$ to input to the extremal feed-forward blocks, and stack $d = 4$ layers. Every layer uses a RELU activation, apart from the last one, which uses the identity function. The log standard deviation scaling factor is 1. The total number of learnable parameters of the proposal distribution is $40,408$. The DMM similarly represents the proposal by a Gaussian. Instead of a 1D convolution, the DMM parameterises the mean and log covariance by assimilating the results of foward and backwards in time LSTMs (Hochreiter & Schmidhuber, 1997); full details can be found in (Krishnan et al., 2017). The total number of parameters belonging to the DMM proposal is $9,384$.

The TCVAE architecture is taken exactly from (Acciaio et al., 2024), so we avoid repeating many of the details here. The TCVAE consists of a normalising flow latent-state generative prior, a Gaussian proposal, and a deterministic observation model. The normalising flow prior has three stacked RealNVP layers with a hidden dimension of 250. The observation and proposal distributions consist of forward-in-time (that is, causal) stacked LSTMs with hidden dimensions of 16 and linear layers to project the inputs and outputs of the LTSMs up or down as appropriate. The prior, observation model and proposal have $2,198,880$; $5,896$; and $4,785$ parameters respectively.

For all models we use the Adam optimiser with learning rate $10^{-4}$ and all other parameters at their default values. We train all our models by minimising $\frac{-1}{120}$ELBO with the appropriate ELBO for the experiment. We propose 32 particles per time-step for 120 time-steps during training. We train for 100 epochs. In unsupervised training, there is no ground truth latent state to which we can ground the latent state representation. Therefore the models in this experiment are at risk of posterior collapse, where the system gets stuck in a local minimum where the proposal is matched to the posterior distribution under the data generating model at the expense of discarding the observations. To mitigate this possibility we use the strategy of Beta-VAEs (Higgins et al., 2017), where we artificially suppress the gradient signal due to the prior and dynamical models. For PVMC, DMM and P-VAE we decrease the suppression from a factor of 0.05 to 1 over the course of training. For Soft DPF it is unclear how one would implement such suppression and it is, to the best of our knowledge, not used in prior work in DPFs. For TCVAE we adopt the constant suppression factor used in (Acciaio et al., 2024). To combat over-fitting we periodically inspect the distribution of returns during training and choose the qualitatively best performing

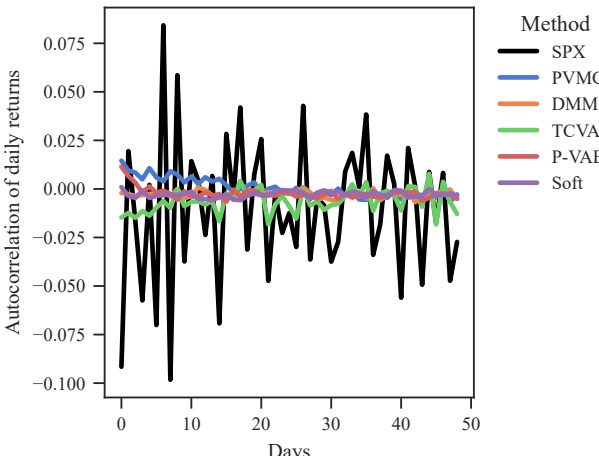

*Figure 7.* Comparison of the mean autocorrelation of daily returns over 1000 generated trajectories of 360 days to 6 non-overlapping real SPX trajectories of the same length.

model.

We supplement the results in the main text with a number of additional plots. Figure 7 shows that all models achieve a small autocorrelation of daily returns (with respect to the noise in the training data). We also plot the histograms of the individual daily returns generated by each method in Figure 8. All methods predict distributions that are overly wide around the peak compared to the training data. Interestingly, this phenomenon is less pronounced in the methods Soft DPF and DMM, which failed to learn the stylised features of the SPX. This suggests that they have collapsed to a local minima where the marginal distributions of returns is accurately learned, but not the distribution on path space.

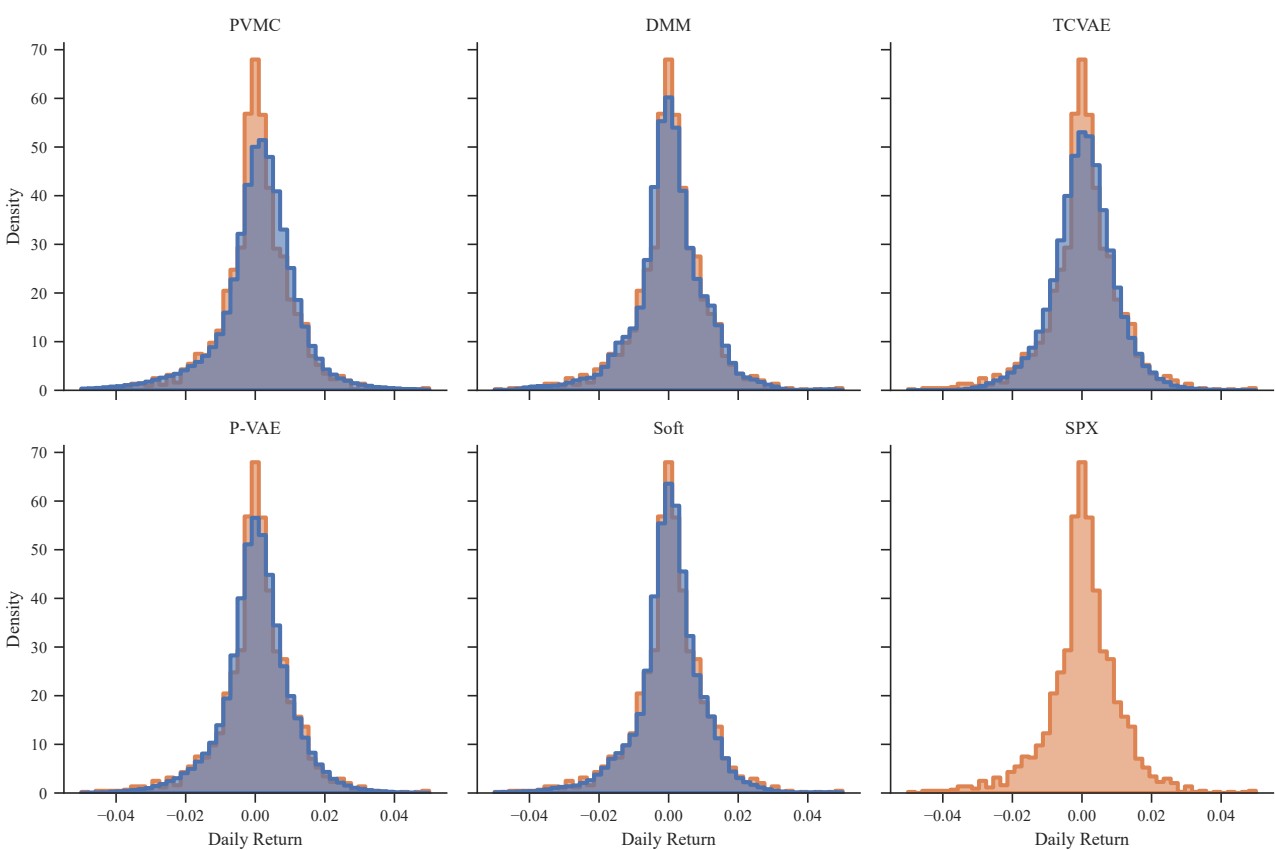

*Figure 8.* Histograms to compare the frequencies of the daily returns between each method and the SPX. The SPX's distribution plotted in orange and the generated data's in blue.

