# OpenReview forum: "Efficient Learning of Deep State Space Models via Importance Smoothing"
_ICML.cc/2026/Conference — ICML 2026 regular_

### Official Review · Reviewer_7knL · 2026-03-05

**Soundness:** 4
**Presentation:** 3
**Significance:** 3
**Originality:** 4
**Overall Recommendation:** 5
**Confidence:** 4

**Summary:**

This paper introduces a new framework for learning a deep state-space model under the hood of particle Monte Carlo. The cimportant ontributions are that it is time-parallel and differentiable, so it addresses the computation and differentiability problems. Moreover, the proposed framework lies in a middle point between two other popular approaches for training a deep SSM, framing a nice story telling.

**Compliance With Llm Reviewing Policy:**

Affirmed.

**Final Justification:**

The authors have addressed all my concerns. I think, at least from the methdological level, this paper does deliver a solid contribution.

**Key Questions For Authors:**

1. The authors claim "unbiased gradient". Indeed it is an unbiased estimate to the marginal likelihood, but it is not to the log and gradient of the log.

2. Since the differentiable particle filtering has already been explored, what is the technical difficulty of extending them to smoothing? As in, One just runs a differentiable particle filter and then applies a backward simulation smoother (e.g., Algorithm 12.2 of Chopin and Papaspiliopolos 2020 book). Although there is too a weight selection procedure, it can be similarly addressed by differentiable categorical sampling. Therefore, I am not very convinced with the claimed contribution on "particle smoother".

3. Since the authors claim the contributions for smoothing, why Table 2 computes the filtering not smoothing?

4. Fully factorized variational proposal. Is Equation (13) reasonable? Factorizing it over time means that all the marginals become conditionally independent. We then cannot have a complete smoothing path. In particle smoothing, it is important to sample a complete trajectory. Is it also true that in your experiment in 5.1 you computed the RTS and compare the results also time marginally? I would like to see the comparison over $p(x_{0:T})$ instead of $p(x_t)$ for $t=1,\ldots,T$. How much does your framework reply on the factorized proposal?

5. I like Theorem 3.2. It shows that the proposed PVMC is tighter than any of the VAE based approaches. However, how is it compared to that of the approaches you refer to as the second paradigm in Introduction?

6. Line 384. "We train PVMC by minimising a linear combination of the negative $L^N_{PVMC}$ and the mean squared error between the estimated smoothing mean and the ground truth population densities." I think this is a bit "cheating" the results due to two reasons: 1) The main purpose of your experiment should be showing that your $L^N_{PVMC}$ is a good objective. However, with the added the mean squared error between ... loss, I am not certain if it is indeed $L^N_{PVMC}$ that is useful. 2) In reality, we never access the latent state but only the measurements. This loss cannot be implemented in practice. Can you train the model only using $L^N_{PVMC}$?


## Minor comments

7. Third paragraph of Introduction. I don't understand the explanation of the first paradigms. I cannot connect VAE and SSMs and thus don't understand the role of a VAE when applied to train a DSSM. Or do you just refer to the variational (lower bound) approach? Is this related to Bartosh et al., 2025? To me, the major difference between the two paradigms is their objectives: log likelihood or a lower bound of it. Or do you mean something more?

Grigory Bartosh, Dmitry Vetrov, Christian A. Naesseth. SDE Matching: Scalable and Simulation-Free Training of Latent Stochastic Differential Equations. 2025

8. Line 114. "The" -> "Then"?

9. Why is $L_{P-VAE}^N = L_{VAE}^N$? Isn't this true only when $N=1$?

10. Section 5.1 What is e_x in Table 1? You should not only compute the error of the smoothing mean. You should also incorporate the smoothing covarian

11. Equation (33b). Isn't 1 + 4 in the denominator trivially 5?

12. Line 363. It doesn't make sense to say dW is a Brownian motion. It is W that is a Brownian motion.

**Limitations:**

No, I don't see a discussion of limitation in the paper (main body).

**Strengths And Weaknesses:**

Soundness:

I think the paper is technically sound. The proposed method is technically correct, and is also formulated correctly. I did not see obvious errors (see Comments).

Presentation: In my opinion it is clear enough. However, like in most parallel algorithm papers, the proposed algorithm is hard to parse and to implement immediately.

Significance: The paper produces a new and useful method for learning a deep state space model, and the results also show that it is indeed computationally efficient and produce good smoothing estimates.

Originality: There were not too much results on parallel-in-time particle smoothing. This paper idea is original.

---

> ### Author Rebuttal · Authors · 2026-03-31
>
> Many thanks for your thorough and constructive feedback.
>
> **1. Unbiased gradients**
> Thanks! Indeed, as stated in Proposition 3.1, $\hat{L}^{N}$ is an unbiased estimator of the likelihood $p(y_{0:T})$, and it does not give an unbiased estimator of $\log p(y_{0:T})$. Per Section 3.2, the PVMC ELBO $\mathcal{E}[\log \hat{L}^{N}] \leq \log p(y_{0:T})$.
> The method returns an unbiased estimator of the gradient of the PVMC ELBO $\mathcal{L}^{N}_{\mathrm{PVMC}}$, not of the gradient of the log marginal likelihood itself. We will clarify this in the revision.
>
> **2. FFBS**
>  FFBS is difficult to parallelise and introduces repeated discrete ancestor selection operations which are a primary source of difficulty in DSMC. For instance, Younis and Sudderth (2024) report that an FFBSS-based learned smoother performs worse than MDPS.
> In an FFBSS-based differentiable smoothing algorithm, these issues would appear not only in the forward filter but again in the backward sampling stage. In typical DPF training some gradient paths avoid resampling for all parameters. This will not be the case in an FBSS where some parameters will have all gradient paths through a long chain of selection steps.
>
> **3. Table 2**
> The first MSE column reports the state estimation error produced by each method’s own inference procedure (in-task evaluation, where inference error corresponds to supervised contribution to the training loss). Thus, for PVMC and MDPS, this is a smoothing MSE.
> “Filtering MSE’’ reports vanilla bootstrap particle filter performance with the learned SSM (out-of-task evaluation, testing whether generalisation beyond the specific inference procedure used during training).
> We will clearly delineate in-task and out-of-task evaluation in the revision.
>
> **4. Factorized proposals**
> Indeed, removing temporal dependence in the proposal does not allow directly representing joint smoothing trajectories $p(x_{0:T}| y_{0:T})$. Full smoothing would be desirable but challenging due to the growing dimension of the target, and so full smoothers typically rely on a non-differentiable selection procedure.
>
> Importantly, PVMC is not inherently reliant on a factorised proposal (see Appendix E.2). We adopt a factorised proposal in our experiments for two principal reasons. Firstly, it is naturally parallel-in-time. Secondly, we can guarantee that the mean state and likelihood estimates are improved with the number of particles at the usual MC rate of $\frac{1}{N}$.
> In our revision we will clarify that factorised proposals are a design choice and not fundamentally required.
>
> **5. Context of Theorem 3.2**
> Theorem 3.2 compares PVMC and VAE-based objectives under the same proposal. Comparison with DSMC ELBOs is difficult, because DSMC methods introduce specific proposal distributions (see the previous question).  We add an explicit explanation of the difficulty in comparing $\mathcal{L}^{N}_{\mathrm{PVMC}}$ to DSMC ELBOs in our paper.
>
> **6. Line 384. Supervised vs unsupervised losses:**
> Thanks! That specific experiment evaluates whether PVMC can be used effectively in the standard supervised DSMC training setting (unsupervised losses are evaluated in Section 5.1 and 5.3), where “ground truth” data are expensive calibration data, and the observations are less well resolved cheap measurements. This situation is common in, for example, machine learning approaches to localisation, such as visual localisation (Sarlin *et al.* (2023), *OrienterNet: Visual Localization in 2D Public Maps with Neural Matching*) as well as time series arising in industrial processes.
>
> We will include pointers to the DSMC literature that follow this approach, including: Jonschkowski _et al._ (2018), and Younis and Sudderth (2025).
>
> **7. VAEs and SSMs**
> It is common in both VAE-SSMs and DSMC to target ELBOs.
> VAE-SSMs are special cases of the standard VAE where the generative model is given an SSM structure. Bartosh *et. al* (2025) study a highly related problem where the SSM is generated by an SDE. In the VAE-SSM paradigm the proposal and generative distributions may or may not share parameters, but crucially the proposed trajectories are independent, whereas in DSMC the trajectories come from a particle filter.
>
> **8. 11. 12. Errors**
> Thanks, these will be corrected!
>
> **9. ELBO equality**
> We have that $\mathcal{L}^{N}_{\mathrm{P-VAE}}$ is an empirical average of expected trajectory log-likelihoods.
>
> This is the same as $\mathcal{L}^{N}_{\mathrm{VAE}}$. Thus they are equivalent for any choice of $N$. This is shown explicitly in Equation (58).
>
> **10. Table 1**
> Thanks, $e_{x}$ is the average L2 error between the estimated and exact RTS smoother posterior means. To assess posterior fidelity, we will report the sliced 2-Wasserstein distance between particle approximation and exact posterior at the central time-step ($t=250$).
> |Method|Sliced 2-Wasserstein distance|
> |-|-|
> |PVMC (learned proposal)| **0.048**|
> |PVMC (Kalman proposal)|0.065|
> |TFS|1.15|
> |d-SMC|0.19|

---

> > ### Author Rebuttal · Reviewer_7knL · 2026-04-02
> >
> > Thank you for your replies! I will consider increasing my score as you have addressed most of my concerns. However, I now have spotted a reproducibility issue.
> >
> > We are now working on a similar problem and are trying to run your code given in the supplementary material, and found that:
> >
> > - missing requirements.txt.
> > - some scripts were not able to run. E.g., running linear_gaussian/main.py gives error "ImportError: cannot import name 'NegativeKernelLogLikelihood' from 'smoother_outputs' ". I cannot find this class in any file in this project.
> > - the implementation of diffusion_DPF.py seems to be wrong. Lines 108-120 do not implement the original algorithm and look like a non-trivial modification. I think the authors should explain this. We tried to also implement it for the same Volterra model, but we got lower results in terms of MSE and Filtering MSE compared to yours.
> > - None of the scripts in `bearings_only` is runable.
> >
> > But I still appreciate the authors providing the code. This makes the proposed algorithm clearer to understand.
> >
> > I also have one more suggestion. The current presentation of the paper, like the authors explained to other reviewers, is too tied to the SMC community. I will not have problem if you submit the paper to a statistics journal. But **ICML is a broad community**, and by its nature, you should make your paper accessible to a broad audience. I kindly ask the authors to revise the paper in this regard.

---

> > > ### Author Response · Authors · 2026-04-07
> > >
> > > Thank you again for your thorough review, and checking our paper and code in details. We have made all necessary changes to our GitHub repository and we include the link to an anonymised repository here: https://anonymous.4open.science/r/parallel-variational-sequential-monte-carlo-57C8/
> > >
> > > **requirements.txt**
> > >
> > > Thank you for flagging this. We have fixed this in the GitHub repository that we link anonymously. We have also noticed that the raw data for the SPX example is missing from the supplementary material we have included it in the update anonymous repository.
> > >
> > > **running linear_gaussian**
> > >
> > > Thank you for catching this error. We do not use the class NegativeKernelLogLikelihood in any of our experiments. We have removed references to it in the updated anonymous repository.
> > > **diffusion_DPF.py**
> > >
> > > Thank you for checking our implementation. Our implementation of the diffusion DPF is a PyTorch reimplementation of the function “diffusion_resampling()” in resampling.py in the GitHub repository accompanying Anderson and Zhao (2025), https://github.com/zgbkdlm/diffres/blob/main/diffres/resampling.py. During our baseline validation, we observed that their algorithm was unstable, so we introduced a small modification for numerical and statistical stability.
> > > Following your comment, we have implemented Algorithm 3 in Anderson and Zhao (2025) directly following the paper’s pseudocode. On the runs completed so far, this implementation performs better on the Lotka-Volterra experiment. These results are still preliminary because the full set of repeats has not yet completed. We will update the full results in the revision, reporting both for transparency, and clarify exactly which implementation is used in each reported result. We will also document this distinction explicitly in the code release.
> > >
> > > Preliminary results for the direct Algorithm 3 implementation are:
> > >
> > > | MSE | Filtering MSE| 2-SWD|
> > > |---|---|---|
> > > |0.541|0.438|11.6|
> > >
> > > **bearings only**
> > >
> > > The bearings only experiment setting was used primarily to validate our implementation of MDPS and to debug an early version of PVMC. We do not use it in any results we present in the paper and should not have been included in the supplementary material. We appreciate your careful examination and have removed it in the updated anonymous repository.
> > >
> > > **SMC presentation**
> > >
> > > Thank you for your point about accessibility. In the revision, we will revise the presentation to better target a broad ICML audience. Specifically:
> > >
> > > 1.	We will define the slice notation, $x_{a:b}$, explicitly as "\\{x_{t}\\}_{t\in[a,\ldots,b]}" (note: we couldn't get openreview to parse this latex correctly).
> > >
> > > 2.	As suggested by Reviewer RsuC, we will use the message passing description of our algorithm. We will refer to message passing in our introduction when introducing our contribution, in the “Associative prefix sums” paragraph of Section 2, and when describing our algorithm in Section 3.2.

---

### Official Review · Reviewer_WugS · 2026-03-11

**Soundness:** 4
**Presentation:** 3
**Significance:** 3
**Originality:** 3
**Overall Recommendation:** 5
**Confidence:** 2

**Summary:**

This paper introduces Parallel Variational Monte Carlo (PVMC), a new training objective for deep state space models (SSMs). The central idea is to avoid particle filtering entirely while still capturing smoothing-style dependencies across time. Instead of sequential resampling, the method defines an independent proposal distribution over latent states at each time step. Despite this independence assumption, the authors construct a tighter variational lower bound than the standard VAE objective by importance-weighting over all possible trajectories formed from the per-time particles.

Naively, this corresponds to an exponential sum over N^{ T+1} trajectory combinations. The main technical contribution is showing that this combinatorial estimator can be computed efficiently using structured matrix contractions and parallel prefix/suffix scans, reducing the computation to polynomial work and logarithmic parallel span. The resulting PVMC ELBO is provably tighter than both VAE and IWAE objectives in this setting.

**Compliance With Llm Reviewing Policy:**

Affirmed.

**Key Questions For Authors:**

_

**Limitations:**

yes

**Strengths And Weaknesses:**

- Strengths:
  - The use of associative matrix contractions and parallel scan to evaluate a combinatorial smoothing objective is elegant and technically nontrivial.
  - The ELBO hierarchy and equivalence results (e.g., P-VAE = VAE) are clean and well-motivated.
  - Deep state-space models remain an important modeling class in time-series and control applications.

- Weaknesses:
  - The independence assumption in the proposal may limit performance in regimes with strong smoothing correlations, near-deterministic dynamics, or long time horizons where temporal consistency is critical. Structured proposals (e.g., SMC-style) may be more robust in such settings. The paper would benefit from experiments probing these regimes.
  - Although the parallel span is logarithmic in T and N, the total computational work scales cubically in the number of particles due to repeated N×N matrix multiplications, which may limit the practical particle count.

---

> ### Author Rebuttal · Authors · 2026-03-31
>
> Many thanks for your positive and helpful comments. We respond to the weaknesses below:
>
> **1. The independence assumption in the proposal may limit performance in regimes with strong smoothing correlations, near-deterministic dynamics, or long time horizons where temporal consistency is critical. Structured proposals (e.g., SMC-style) may be more robust in such settings. The paper would benefit from experiments probing these regimes.**
>
> We agree that a factorised proposal may perform less well in certain regimes than a structured proposal such as those developed by Krishnan *et al* (2017) *Structured inference networks for non-linear state space models*; or the standard SMC proposal process. Hence, we provide recipes in Appendix E.2. for adapting PVMC to use both of these proposals.
>
> The trade-off is that these more general proposal families fall outside the scope of our current theoretical analysis. In particular, the theorems results establishing that the MSE of our the proposed estimators converge to zero with rate $\frac{1}{N}$ would not be straightforward to establish.  Extending the theory to cover broader proposal classes for which analogous theoretical guarantees can be given is an interesting direction for future work. Theorem 3.2, the ELBO hierarchy, does not depend on the independence assumption. We will clarify this in the revision.
>
> We also agree that further empirical validation in these regimes would be valuable. However, a systematic comparison across proposal distributions would substantially broaden the scope of the paper. Instead, in the revision, we will explicitly delineate the regimes in which PVMC is expected to perform less well than structured proposals.
>
> **2. Although the parallel span is logarithmic in T and N, the total computational work scales cubically in the number of particles due to repeated N×N matrix multiplications, which may limit the practical particle count.**
>
> You are correct that the total computational cost of PVMC is high, and that the method is only feasible on highly parallel hardware. However, PVMC is primarily intended as a training-time method for learning SSMs rather than as a deployment-time inference algorithm. In this setting, the higher computational cost can be amortised by modern parallel hardware during training, while the learned model can subsequently be used with significantly cheaper inference schemes. Indeed, in Sections 5.2 and 5.3, we demonstrate that models trained with PVMC generalise well to downstream tasks such as filtering and data generation, supporting this amortised inference perspective.
>
> Moreover, DSMC algorithms are typically trained with relatively small particle counts compared to classical SMC, often with $N<128$. This is motivated not only for computational benefit but also statistical considerations. It is known that for importance weighted VAEs increasing $N$ can reduce the signal-to-noise ratio of the gradients with respect to the proposal network parameters Rainforth *et al.* (2018) *Tighter Variational Bounds are Not Necessarily Better*. For this reason, we elect to use fewer particles than our hardware would allow in our experiments.

---

> > ### Author Rebuttal · Reviewer_WugS · 2026-04-07
> >
> > Thank you for the response to the review, I will retain my score.

---

### Official Review · Reviewer_RsuC · 2026-03-13

**Soundness:** 2
**Presentation:** 3
**Significance:** 4
**Originality:** 4
**Overall Recommendation:** 4
**Confidence:** 3

**Summary:**

The paper develops an algorithm based on sequential importance sampling, variational inference, and associative prefix sums to perform timewise marginal smoothing in probabilistic state-space models.  The resulting lower bound on the log marginal likelihood via the expectation of log importance weights is proved to be tighter than the standard ELBO and than IWAE bounds that make no use of the temporal structure to smooth over more de facto particles. Experiments show that the resulting variational and SMC combination beats other approximate smoothing algorithms on linear-Gaussian state-space models, a stochastic Lotka-Volterra predator-prey dynamics, and generative modeling of the S&P 500, demonstrating the effectiveness of PVMC.

**Compliance With Llm Reviewing Policy:**

Affirmed.

**Key Questions For Authors:**

Can the authors describe how PVMC is optimized, use standard machine-learning (rather than pure SMC) notation, and at least reference Algorithms D.3 and D.4 in the main text?

**Limitations:**

yes

**Strengths And Weaknesses:**

Strengths:
* Strong empirical results on a decent variety of experimental settings
* Strong grounding in importance-weighting methods and variational inference fundamentals

Weaknesses:
* Nonstandard notation for the subfield, bearing more resemblance to "An Introduction to Sequential Monte Carlo" than to typical ICML, ICLR, or NeurIPS papers applying SMC and variational inference methods.
* Related to the suppressing of variational parameters from notation, no gradient estimator for the PVMC ELBO is given, nor is a loss for the differentiable SMC resampling steps.
* Proofs are left to appendices.

---

> ### Author Rebuttal · Authors · 2026-03-31
>
> Many thanks for your positive review and constructive feedback.
>
> **1. Nonstandard notation for the subfield, bearing more resemblance to "An Introduction to Sequential Monte Carlo" than to typical ICML, ICLR, or NeurIPS papers applying SMC and variational inference methods. Can the authors describe how PVMC is optimized, use standard machine-learning (rather than pure SMC) notation, and at least reference Algorithms D.3 and D.4 in the main text?**
>
> Thank you for raising this point and your question. Our goal is to make the paper accessible to both the SMC and variational inference communities. In the revision, we will address the point by (i) adding an alternative VAE-description of PVMC as an Appendix and reference it in the main text; (ii)  make the optimisation setup explicit, including parameter dependence and gradient estimation; and (iii) add references in the main text to Algorithms D.3 and D.4.
>
> We include below a description of our algorithm in VAE language.
>
> When used to target an ELBO, $\mathcal{L}^{N}_{\mathrm{PVMC}}$, PVMC can be framed as a variational auto-encoder on sequences.
>
> The observed data, $y_{0:T}$, and the latent state, $x_{0:T}$, are sequences of $T+1$ random variables.
>
> To aid the learning of complex dynamics we impose structural priors on the data generating model. Specifically, the prior distribution of $x_{0:T}$ is assumed to be a Markov process, $x_{t>0} \sim M_{t}(\cdot| x_{t-1})$. With $x_{0}$ drawn from some initialisation distribution $P$. The data is generated conditionally independently from the data at any other time-step by sampling the observation model, $y_{t} \sim H_{t}(\cdot| x_t)$. This is analogous to the decoder network in a VAE. Thus, the generative process mimics the structure of a state space model. We note that frequently in classical VAEs the observation model is restricted to be deterministic, whereas we allow the observations to be random conditional on the latent state.
>
> We have defined a structured generative model, however like in the classical VAE setting, the posterior distribution $p(x_{0:T}\mid y_{0:T})$ is not tractable. Therefore, we learn a variational approximation $V_{0:T}(x_{0:T} \mid y_{0:T})$. This is analogous to the encoder network of a VAE.
>
> It is well known that the variational bound (the ELBO) can be made tighter by averaging over several independently proposed particles given a single datum, $y_{0:T}$. This approach is highly related to importance sampling Burda *et al.* (2016) *Importance weighted autoencoders*.
>
> In PVMC we propose $N$ particles per time-step. A classic importance weighted auto-encoder would be restricted to choosing $N$ independent trajectories. In PVMC however, we consider every possible path through the set of particles, for a total of $N^{T+1}$ possible trajectories. The resulting variational lower bound is provably tighter than the standard importance weighted auto-encoder (Theorem 3.2.). To make calculation of this ELBO feasible we employ a parallelisable message passing approach making use of the associativity of matrix multiplication. We refer readers Algorithms D.3 and D.4 and Theorem 3.1 for detail.
>
> In addition to obtaining the sum of the importance weights, and therefore the ELBO, we can additionally compute the per-time-step importance weights. Thereby obtaining an approximation to the marginal smoothing distribution.
>
> **2. Related to the suppressing of variational parameters from notation, no gradient estimator for the PVMC ELBO is given, nor is a loss for the differentiable SMC resampling steps.**
>
> Regarding the current presentation suppressing the dependence on model and variational parameters, we will make this dependence explicit in the revision when first introducing the objective, e.g. by writing  $\mathcal{L}^{N}_{\mathrm{PVMC}}(\theta,\phi)$, and only suppressing $(\theta,\phi)$ later for readability.
>
> We will also clarify the gradient estimator used to optimise the PVMC ELBO. In our method, the stochastic operations are pathwise / reparameterised, and the ELBO computation itself is otherwise deterministic, so gradients are obtained by automatic differentiation through the resulting computation graph.
>
> Unlike DSMC methods, PVMC does not rely on sequential resampling steps, and therefore does not require a separate surrogate loss for differentiable resampling. We will make this contrast explicit in the revision.
>
> **3. Proofs are left to appendices.**
> We agree that leaving all proofs to the appendix makes the main ideas harder to parse. In the revision, we will add brief proof intuition / proof sketches for the main results in the main text, while keeping the full technical proofs in the appendices.

---

> > ### Author Rebuttal · Reviewer_RsuC · 2026-04-04
> >
> > Thank you for clarifying the notation, the gradient estimator, and the proof sketches.  I assure you, you do not need to explain what a probabilistic state-space model is in some sort of "VAE language" in which likelihoods are not presumed to consist of probability densities!  What would be most helpful would be to emphasize that by use of the associativity of matrix multiplication and message-passing, you can efficiently compute your tighter importance-weighted evidence bound and sample from the smoothing marginals.
> >
> > One last question: if you do not rely on resampling steps, then in what sense do you require differentiable resampling?  Are you using differentiable resampling to refer to something other than the differentiable form of sequential importance resampling?

---

> > > ### Author Response · Authors · 2026-04-07
> > >
> > > Thank you for your clarification and concrete suggestion. You are absolutely right that the key point is to emphasise our core contribution that by exploiting the associativity of matrix multiplication in a message-passing formulation, we can efficiently compute both a tighter importance-weighted evidence bound and the marginal smoothing weights. We will improve the accessibility of our paper by making the connection to message passing in our introduction when introducing our contribution, in the “Associative prefix sums” paragraph of Section 2, and when describing our algorithm in Section 3.2.
> > >
> > >
> > > **One last question: if you do not rely on resampling steps, then in what sense do you require differentiable resampling? Are you using differentiable resampling to refer to something other than the differentiable form of sequential importance resampling?**
> > >
> > > Thank you for raising the question. To answer this directly, PVMC does not require  differentiable resampling. By avoiding sequential resampling, we avoid the need for a surrogate loss or gradient estimator associated with differentiable resampling. All stochastic components in PVMC are fully reparameterised, and the remaining computation of the objective is deterministic and differentiable, so gradients are obtained by ordinary backpropagation through the computation graph. We will revise the paper to make this distinction explicit and to avoid potential confusion.

---

### Official Review · Reviewer_SHy8 · 2026-03-13

**Soundness:** 3
**Presentation:** 2
**Significance:** 3
**Originality:** 2
**Overall Recommendation:** 4
**Confidence:** 3

**Summary:**

The paper proposes a framework for learning the posterior to deep state space models through a variational approach with a factorized proposal distribution that allows for parallel sampling of particles across time. The method defines trajectory importance weights that implicitly account for all possible particle paths through time, and shows that the resulting marginal smoothing weights can be computed efficiently using parallel prefix/suffix scan operations. This culminates in an ELBO that is tighter than several existing objectives such as IWAE. The authors demonstrate this method on two synthetic tasks (assessing how well latent states are estimated) and one real-world time-series dataset (roughly assessing goodness of fit) against 5 baseline methods, showcasing superior performance with the proposed method.

**Compliance With Llm Reviewing Policy:**

Affirmed.

**Final Justification:**

I will maintain my initial score. Should the edits made by the authors follow through with what they said then I hope to see this paper be accepted.

**Key Questions For Authors:**

1. How does the memory consumption of the associative scan scale in practice as the number of particles N increases? Is there a point where the memory cost outweighs the speed benefits of parallelism?
2. The paper assumes a fully factorizable variational proposal. How sensitive is PVMC to the quality of this proposal compared to traditional particle filters that use the transition prior?
3. How does the method handle very long sequences (T>1000)? Does the smoothing approach suffer from "particle degeneracy" over long horizons, and if so, how does PVMC mitigate this without sequential resampling?

**Limitations:**

yes

**Strengths And Weaknesses:**

Soundness:
+ The method is based on a principled importance sampling formulation for state space models and appears technically well grounded.
+ The paper provides theoretical analysis, including unbiased likelihood estimation and standard Monte Carlo convergence guarantees.
+ Experiments include both controlled synthetic settings and a real-world dataset with comparisons to several relevant baselines.
− The factorized proposal assumption removes temporal dependence and may limit proposal quality in complex dynamics.
− The paper provides limited analysis of importance weight variance or behavior for long sequences.

Presentation:
+ The paper is generally well structured and clearly motivates the goal of bridging variational DSSMs and differentiable SMC methods.
+ Figures and diagrams help illustrate the algorithmic structure and differences from prior approaches.
− Some technical sections, particularly the scan-based weight computation, are dense and difficult to follow.
− Providing pseudocode or a clearer step-by-step description of the algorithm would improve clarity.

Significance:
+ The work addresses scalable and principled learning for deep state space models. Enabling parallel-in-time smoothing could improve the practicality of particle-based inference methods on modern hardware.
− Empirical validation is somewhat limited, primarily focusing on synthetic tasks and a single real-world dataset.
− It is unclear how well the method scales to longer sequences or more complex real-world problems.

Originality
+ The paper introduces a novel formulation that enables parallel computation of smoothing marginals via associative scan operations.
+ The proposed PVMC objective provides a new variational bound connected to existing objectives such as IWAE.
− The method builds on existing ideas from importance-weighted variational inference and differentiable particle filtering, so some components are incremental.
Missing related works:
- Johnson, Matthew J., et al. "Composing graphical models with neural networks for structured representations and fast inference." Advances in neural information processing systems 29 (2016).
- Zhao, Yixiu, and Scott Linderman. "Revisiting structured variational autoencoders." International Conference on Machine Learning. PMLR, 2023.

---

> ### Author Rebuttal · Authors · 2026-03-31
>
> Many thanks for your detailed and constructive feedback.
>
> **Significance – Limited evaluation**
>
> We chose settings that are challenging for previous approaches. The Lotka-Volterra model in Section 5.2 was shown by Anderson and Zhao (2025) *Diffusion differentiable resampling* to be challenging for all tested DSMC algorithms including their proposed approach. The generative modelling of financial time-series is an important and actively studied problem (Meldrum *et al.* (2025) *New Money: A Systematic Review of Synthetic Data Generation for Finance*). On this task, we compare against a state-of-the-art approach proposed specifically for SPX generation, TCVAE.
>
> **Originality: Missing related works**
>
> Thanks! We will add both references in the revision and clarify how they relate to our contribution. Johnson *et al.* (2016) introduces structured variational autoencoders as a general framework combining graphical model priors, neural network observation models to approximate the posteriors via variational message-passing inference. Zhao and Linderman (2023) revisited this line of research with a parallelised implementation. Both works exploit Kalman smoothing for efficient inference of a surrogate variational inference problem restricted to linear and Gaussian priors and conjugate Gaussian potentials.
>
> These works are closely related to ours, but our contribution is different. PVMC develops a particle-based parallel smoothing objective for more general state space models, without relying on linear Gaussian structure.  We will revise the related work discussion to make both the differences explicit.
>
> **1. Memory consumption**
>
> As with most parallel algorithms, PVMC trades greater parallelism for higher memory usage. The memory cost of training using the ELBO, $\mathcal{L}^{N}_{\text{PVMC}}$, scales as $\mathcal{O}(N^{2}T)$, and the smoothing estimates as $\mathcal{O}(N^{2}T\log T)$. We compare this to the differentiable baseline approaches below:
>
> |Method|Training memory|
> |---|---|
> |PVMC (smoothing)|$\mathcal{O}(N^{2}T\log T)$|
> |PVMC (ELBO)|$\mathcal{O}(N^{2}T)$|
> |Soft DPF|$\mathcal{O}(NT)$|
> |Stop-grad DPF |$\mathcal{O}(N^{2}T)$|
> |Diffusion DPF|$\mathcal{O}(N^{2}T)$|
> |MDPS|$\mathcal{O}(N^{2}T)$|
> |DMM|$\mathcal{O}(NT)$|
> |TCVAE|$\mathcal{O}(NT)$|
>
> When computing the ELBO, PVMC has the same asymptotic memory complexity as the Stop-grad DPF, Diffusion DPF, and MDPS, and is only more expensive by a factor of $\log T$ for smoothing estimates.
> In practice memory constraints will limit the feasible $N$. An interesting direction for future work would be to study hybrid sequential-parallel recursions, where several partial sequences are smoothed sequentially but multiple such blocks are handled in parallel. Then, the partial sequences are combined via an associative operator.
>
> **2. Factorised proposal**
>
> Our choice to use a factorisable proposal in our experiments is a design choice rather than a requirement of PVMC. In appendix E.2. we outline a recipe for designing an unbiased Markov-sequential proposal. However, without factorisation establishing $\frac{1}{N}$ MSE convergence guarantees becomes significantly more challenging. Expanding the class of proposals for which similar theoretical guarantees can be given is an interesting direction for future research. Theorem 3.2, the ELBO hierarchy, does not depend on the independence assumption, we will clarify this in the revision.
>
> A systematic empirical comparison of proposal parameterisations would have substantially expanded the scope of this paper. Instead, we will include discussion carefully delineating the regimes where we expect a factorisable proposal to be robust.
> Finally, we note that PVMC allows learning non-bootstrap proposals with unbiased gradient estimators, which is a key advantage over many DSMC methods that lack strong empirical or theoretical support for learning complex proposals.
>
> **3. Long sequences**
>
> PVMC is not susceptible to particle degeneracy in the same sense as classical SMC methods. In classical SMC algorithms, particle and weight degeneracy arise because the proposal distribution only has access to past information, whereas in PVMC, the proposal can depend on the full observation sequence.
> Since DSMC algorithms generally suffer from gradient instability for longer sequences, sequence lengths are often short. The horizons studied in our paper are already long in comparison to previous work. For reference, we report the length of the longest training sequence used in each paper that proposed the baseline algorithms that learn an SSM.
>
> |Method|Longest $T$|
> |---|---|
> | Our paper | 500 |
> | Soft DPF | 24 |
> | Stop-Grad DPF | 200 |
> | Diffusion DPF | 256 |
> | MDPS | 100 |
> | DMM | 25 |
>
> Finally, in Section 5.3 we demonstrate that models trained with PVMC can generalise beyond the training horizon. We train on 128-step windows of SPX data and generate trajectories of length 360. We will clarify this point in the revision.

---

> > ### Author Rebuttal · Reviewer_SHy8 · 2026-04-02
> >
> > I thank the authors for their response. I will maintain my original score.

---

### Decision · Program_Chairs · 2026-04-30

**Decision:**

Accept (regular)

**Comment:**

Two reviewers recommend Accept, and two reviewers recommend Weak Accept. Reviews contain a list of issues which were sufficiently addressed by the authors at the discussion stage. I therefore recommend Accept.